# Function Encoders:
# A Principled Approach to Transfer Learning in Hilbert Spaces

**Tyler Ingebrand** [1 2]   **Adam J. Thorpe** [1]   **Ufuk Topcu** [1 3]

## Abstract

A central challenge in transfer learning is designing algorithms that can quickly adapt and generalize to new tasks without retraining. Yet, the conditions of when and how algorithms can effectively transfer to new tasks is poorly characterized. We introduce a geometric characterization of transfer in Hilbert spaces and define three types of inductive transfer: interpolation within the convex hull, extrapolation to the linear span, and extrapolation outside the span. We propose a method grounded in the theory of function encoders to achieve all three types of transfer. Specifically, we introduce a novel training scheme for function encoders using least-squares optimization, prove a universal approximation theorem for function encoders, and provide a comprehensive comparison with existing approaches such as transformers and meta-learning on four diverse benchmarks. Our experiments demonstrate that the function encoder outperforms state-of-the-art methods on four benchmark tasks and on all three types of transfer.

**Project page**: tyler-ingebrand.github.io/FEtransfer

## 1. Introduction

Learned models must be able to draw upon multiple knowledge sources to handle tasks that were not encountered during training. For example, robots operating in remote, unstructured environments must be able to adapt to unseen

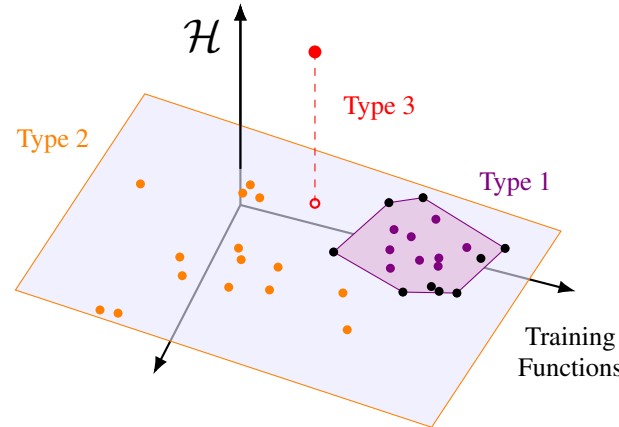

*Figure 1.* **The Categorization of Transfer Learning.** Black points are functions present in the training set. Purple points indicate type 1 transfer, *interpolation within the convex hull*. Orange points represent type 2 transfer, *extrapolation to the linear span*. The red point is type 3 transfer, *extrapolation to the Hilbert space*.

scenarios or terrain, and computer vision models for medical diagnoses should be able to generalize to entirely new tasks, such as identifying emerging or rare diseases. While training on internet-scale data has shown promise in improving generalization performance by increasing data diversity, it does not avoid the fundamental issue of gaps in the data, does not directly address knowledge transfer, and does not innately enable adaptation to new, evolving tasks. Thus, a crucial goal is to develop models that are capable of transfer at runtime without retraining. Addressing these challenges requires incorporating mathematical structure with learning-based representations to quantify and exploit the relatedness of tasks seen during training.

We consider an inductive transfer setting and present a geometric characterization of inductive transfer using principles from functional analysis. Inductive transfer involves transferring knowledge to new, unseen tasks while keeping the data distribution the same. For instance, labeling images according to a new, previously unknown class, where only a few examples are provided after training. While prior works have studied transfer learning, gaps remain in identifying when learned models will succeed and when they will fail. We seek a characterization of inductive trans-

---

[1]Oden Institute for Computational Engineering and Sciences, The University of Texas at Austin, Austin, TX, USA [2]Chandra Department of Electrical and Computer Engineering, The University of Texas at Austin, Austin, TX, USA [3]Department of Aerospace Engineering and Engineering Mechanics, The University of Texas at Austin, Austin, TX, USA. Correspondence to: Tyler Ingebrand <tyleringebrand@utexas.edu>.

*Proceedings of the 42nd International Conference on Machine Learning*, Vancouver, Canada. PMLR 267, 2025. Copyright 2025 by the author(s).

fer, based on Hilbert spaces, which will provide intuition about the difficulty of a given transfer learning problem. Formally, we consider a Hilbert space $\mathcal{H}$ of tasks, and the inductive transfer problem is to accurately represent a new, unknown task $f \in \mathcal{H}$ without retraining, using only a few data examples provided at runtime.

Specifically, we characterize transfer using three types:

(Type 1) *Interpolation within the convex hull.* Tasks that can be represented as a convex combination of observed source tasks.

(Type 2) *Extrapolation to the linear span.* Tasks that are in the linear span of source tasks, which may lie far from observed data but share meaningful features.

(Type 3) *Extrapolation to the Hilbert space.* Tasks that are outside the linear span of the source predictors in an infinite-dimensional function space. Type 3 transfer is the most important and challenging form of transfer.

We propose a novel method to achieve transfer across all three types using the theory of function encoders (Ingebrand et al., 2024b). Function encoders learn a set of neural network basis functions to represent elements in a Hilbert space. We generalize the theory of function encoders to the full Hilbert space transfer learning setting. Further, we show that this approach has a natural and principled means of extrapolation, a difficult task for existing inductive transfer learning approaches.

## 1.1. Related Work

Transfer learning and generalization have been a key focus of the machine learning community. However, much of the existing literature focuses on statistical shifts (out-of-distribution generalization) rather than structural and task-related perspectives (Zhuang et al., 2021). We employ the definitions from Pan & Yang (2010), which distinguishes transfer learning based on differences between the source and target domains. Our focus is on inductive transfer learning. Several transfer learning approaches focus on modeling task relatedness, including via graph-based representations (Eaton et al., 2008) and manifolds (Ko et al., 2024). We consider a similar geometric view of task relatedness which depends on the relationship between a target task and the training data.

Meta-learning approaches, such as MAML (Finn et al., 2017) and related meta-adaptation methods (Nichol et al., 2018), aim to adapt to new tasks by learning parameters or representations that can quickly adapt to new data (Hospedales et al., 2022). These approaches are effective in scenarios with a high degree of similarity between the source and target tasks. However, they require task-specific fine-tuning or retraining, which may fail when tasks are only weakly related or when task relatedness is not explicitly modeled. In contrast, we compute a task representation from data without retraining the underlying model.

Other approaches use the idea of basis functions, but in distinct ways. Kernel methods fit a function by placing a basis function at every observed data point, and represent the function as a linear combination of these basis functions (Hofmann et al., 2008). Kernel methods can also be used for transfer learning (Radhakrishnan et al., 2023). However, kernel methods may scale poorly with data, because the size of the Gram matrix grows with the amount of data, and require prior knowledge, e.g., the user must choose the kernel. In contrast, the function encoder scales well the amount of data because it uses a fixed number of basis functions, and requires no prior knowledge because the basis functions are learned. Dictionary learning decomposes a data matrix into a set of atoms. Functions are represented as a linear combination of the atoms, often with sparse regularization (Tillmann, 2015; Aharon et al., 2006). However, these atoms are only represented at fix input locations, and thus are more akin to a discretized representation of a basis function. In contrast, the function encoder can be evaluated at any point in the input domain.

Internet-scale training and fine-tuning pretrained models is gaining interest for improving generalization performance. For example, the Open-X Embodiment dataset (O'Neill et al., 2024) aggregates data from diverse robot platforms to train models that generalize across tasks and robot systems. These approaches seek to leverage sheer data volume and scale to enable transfer instead of leveraging structural insights or feature relationships between tasks. Transformers (Vaswani et al., 2017) and generative pretrained models perform well on language tasks and time-series prediction, but perform poorly even in simplistic benchmark transfer tasks. There is some preliminary work on assessing transfer quality for pretrained models (Mehra et al., 2024). Our approach may be useful for such models, e.g. in mixture-of-experts (Jacobs et al., 1991).

## 1.2. Contributions

We summarize our contributions as follows.

**A geometric characterization of transfer.** We define and characterize three types of inductive transfer, which capture geometric task relationships in Hilbert spaces.

**A method to achieve transfer across all three types.** We present a principled method for inductive transfer using the theory of function encoders (Ingebrand et al., 2024b). We propose a novel approach to train function encoders using least squares, prove a universal function space approx-

imation theorem for function encoders, and generalize the function encoder to new function spaces.

**A comprehensive comparison with the state of the art.** We compare state-of-the-art approaches for inductive transfer and meta-learning with our proposed approach on several benchmark transfer tasks, including regression, image classification, camera pose estimation, and dynamics modeling. We demonstrate that our approach achieves comparable performance at type 1 transfer and superior performance at type 2 and type 3 transfer.

## 2. Background

### 2.1. Inductive Transfer in Hilbert Spaces

We present the following definitions, adapted from Pan & Yang (2010) of domains, tasks, and transfer learning.

**Definition 1** (Domain). *A domain $\mathcal{D} = (\mathcal{X}, \mathbb{P}(\mathcal{X}))$ consists of an input space $\mathcal{X}$ and a marginal distribution $\mathbb{P}(\mathcal{X})$.*

**Definition 2** (Task). *A task $\mathcal{T} = (\mathcal{Y}, f)$ consists of an output space $\mathcal{Y}$ and a predictor $f : \mathcal{X} \to \mathcal{Y}$, which is not observed, but can be learned from data. Alternatively, the predictor can be a conditional distribution $\mathbb{P}(\mathcal{Y} \mid \mathcal{X})$.*

In general, we presume that we do not have access to the predictors directly. Instead, we presume access to a dataset consisting of pairs $(x, y) \in \mathcal{X} \times \mathcal{Y}$.

**Definition 3** (Dataset). *A dataset $D = \{(x_i, y_i)\}_{i=1}^{m}$ consists of points $x_i$ drawn from $\mathbb{P}(\mathcal{X})$, and the corresponding labels $y_i$ from $\mathcal{Y}$.*

**Definition 4** (Transfer Learning, Pan & Yang, 2010, Definition 1). *Let $\mathcal{D}_S$ and $\mathcal{T}_S$ be the source domain and task, and $\mathcal{D}_T$ and $\mathcal{T}_T$ be the target domain and task. Transfer learning seeks to improve the target predictor $f_T$ using the knowledge in $\mathcal{D}_S$ and $\mathcal{T}_S$, where $\mathcal{D}_S \neq \mathcal{D}_T$ or $\mathcal{T}_S \neq \mathcal{T}_T$.*

Specifically, we focus on an *inductive* transfer scenario, where the source and target predictors differ, meaning $f_S \neq f_T$ or, alternatively, $\mathbb{P}(\mathcal{Y}_S \mid \mathcal{X}_S) \neq \mathbb{P}(\mathcal{Y}_T \mid \mathcal{X}_T)$, but the output spaces $\mathcal{Y}_S = \mathcal{Y}_T$ and domains $\mathcal{D}_S = \mathcal{D}_T$ are the same. For simplicity of notation, we drop the subscripts $S$ and $T$ on $\mathcal{X}$ and $\mathcal{Y}$ where appropriate. Additionally, we presume that multiple source datasets $D_{S_1}, \ldots, D_{S_n}$ are available during training, which is similar to the so-called multi-task learning scenario (Caruana, 1997). The model thus has access to multiple datasets during training, and we seek to transfer the knowledge from the source domain $\mathcal{D}_S$ and tasks $\mathcal{T}_{S_1}, \ldots, \mathcal{T}_{S_n}$ to a new target task $f_T$. This scenario is increasingly common in practice. For instance, in few-shot image classification, we have access to a large dataset of diverse images and their corresponding labels, and we seek to transfer to a new class. In robotics, we have access to data collected from multiple environments,

and we aim to transfer dynamics estimates or policies to a new environment. To make use of Hilbert space theory, we make the following assumption about the tasks.

**Assumption 1.** *Consider a Hilbert space $\mathcal{H}$ of functions from $\mathcal{X}$ to $\mathcal{Y}$ equipped with the inner product $\langle \cdot, \cdot \rangle_{\mathcal{H}}$. We assume that the predictors $f_{S_1}, \ldots, f_{S_n}$ and $f_T$ are elements of $\mathcal{H}$.*

Note that Assumption 1 is not restrictive because Hilbert spaces are generally flexible and encompass a wide number of problems of interest. For instance, under mild regularity assumptions, the space $L^2$ of square-integrable functions is a Hilbert space. We can define Hilbert spaces over many types of functions and tasks, such as probability distributions. See Appendix C for more information.

### 2.2. A Geometric View of Transfer

We first seek to characterize the various geometric relationships between the target task $f_T$ and the source tasks $f_{S_1}, \ldots, f_{S_n}$ via the properties of the Hilbert space. Specifically, we define three types of inductive transfer in Hilbert spaces: interpolation within the convex hull of source predictors, extrapolation to the linear span of source predictors, and extrapolation to the rest of $\mathcal{H}$.

We first consider a typical case of generalization for learned models, where the target predictor lies within the convex hull $C_h$ of the source predictors $f_{S_1}, \ldots, f_{S_n}$,

$$C_h = \left\{ f \in \mathcal{H} \;\middle|\; f = \sum_{i=1}^{n} \alpha_i f_{S_i}, \sum_{i=1}^{n} \alpha_i = 1, \alpha_i \geq 0 \right\}. \tag{1}$$

**Definition 5** (Type 1, Interpolation in the Convex Hull). *Given source predictors $f_{S_1}, \ldots, f_{S_n}$ the target task predictor $f_T$ is in the convex hull $C_h$ of the source predictors.*

Next, we consider extrapolation beyond the convex hull of the source predictors to the linear span,

$$\mathrm{span}\{f_{S_1}, \ldots, f_{S_n}\} = \left\{ f \in \mathcal{H} \;\middle|\; \sum_{i=1}^{n} \alpha_i f_{S_i}, \alpha_i \in \mathbb{R} \right\}. \tag{2}$$

**Definition 6** (Type 2, Extrapolation to the Linear Span). *Given source predictors $f_{S_1}, \ldots, f_{S_n}$ the target task predictor $f_T \in \mathrm{span}\{f_{S_1}, \ldots, f_{S_n}\}$. We presume $f \notin C_h$ to distinguish extrapolation from interpolation.*

Extrapolation to the linear span extends beyond standard generalization performance and tests whether the learned features are meaningful. In other words, extrapolation to the linear span implies that the model has learned something about the structure of the space of predictors.

Lastly, we consider the extreme case where the target task lies outside the span of source predictors. This type of transfer is often the most desirable, since it involves transferring (partial) knowledge to entirely different tasks.

**Definition 7** (Type 3, Extrapolation to $\mathcal{H}$). *The target task predictor $f_T \in \mathcal{H}$, but $f_T \notin \text{span}\{f_{S_1}, \ldots, f_{S_n}\}$.*

Note that this scenario is particularly challenging, since it involves transferring to parts of $\mathcal{H}$ that significantly differ from the observed tasks. Additionally, $\mathcal{H}$ is typically infinite dimensional, while we have finite training tasks, and so there are infinite dimensions on which $f_T$ may differ from the source tasks. It is not possible, a priori, to determine which of these infinite dimensions are relevant.

## 3. Function Encoders

A natural representation of a task $f \in \mathcal{H}$ is via a linear combination of basis functions. We employ a method for learning a finite set of neural network basis functions in Hilbert spaces called the function encoder (Ingebrand et al., 2024b). Function encoders learn a set, $\{g_1, \ldots, g_k\}$, of basis functions parameterized by neural networks to span a Hilbert space of functions. Then, functions in the learned space are represented via a linear combination of the learned basis,

$$f(x) = \sum_{j=1}^{k} c_j g_j(x \mid \theta_j). \tag{3}$$

We first provide a brief overview of the theory of function encoders. Then, we present a novel approach to training function encoders using least-squares optimization, which offers several numerical and computational benefits. Additionally, we present a universal approximation theorem for function encoders. Lastly, we remark that we have generalized all function encoder definitions to use only inner products. Thus by appropriately defining the inner product, we also generalize the algorithm to any function space, e.g., probability distributions for classification.

Function encoders consist of two steps: offline training of the basis functions and online inference.

### 3.1. Offline Training Procedure

The basis functions $\{g_1, \ldots, g_k\}$ are first trained using a set of datasets $\{D_{S_1}, \ldots, D_{S_n}\}$ as in Definition 3 corresponding to the source predictors $f_{S_1}, \ldots, f_{S_n}$. For each function $f_{S_i}$ and dataset $D_{S_i}$, we first compute the coefficients of the basis functions and then obtain an empirical estimate $\hat{f}_{S_i}$ of $f_{S_i}$ via (3).

During training, the coefficients $c$ corresponding to a func-

tion $f$ can be computed using the inner product method,

$$c := \begin{bmatrix} \langle f, g_1 \rangle_{\mathcal{H}} \\ \vdots \\ \langle f, g_k \rangle_{\mathcal{H}} \end{bmatrix}, \tag{4}$$

which leads to a stable learning algorithm (Ingebrand et al., 2024b). Computing (4) involves evaluating the inner product. Many commonly used inner products such as the $L^2$ inner product $\langle f, g \rangle_{L^2} = \int_{\mathcal{X}} f(x)g(x)dx$ involve computing integrals over the input space, which is computationally intractable. Instead, we can use Monte Carlo integration to approximate the inner product using data. For example, we can estimate the $L^2$ inner product using a dataset $D = \{(x_i, y_i)\}_{i=1}^{m}$ as

$$\langle f, g_j \rangle_{L^2} = \int_{\mathcal{X}} f(x)g_j(x)dx \approx \frac{V}{m} \sum_{i=1}^{m} f(x_i)g_j(x_i), \tag{5}$$

where $V$ is the volume of $\mathcal{X}$ and $f(x_i) = y_i$. Since $g_j$ is a neural network basis function, we can query it for arbitrary data points $x_i$. In practice, $V$ is unknown, so we assume $V = 1$, which scales the inner product by a constant value $1/V$. This approximation induces a weighted inner product if $\mathbb{P}(\mathcal{X})$ is not uniform. For a discussion, see Appendix D.

The inner product method of computing the coefficients suffers from two main drawbacks. First, as noted in Ingebrand et al. (2024b), the basis functions naturally tend toward orthonormality during training to minimize the loss. However, during training, the inner product method introduces errors in the coefficient calculations if this property is not explicitly enforced at each step, e.g., using Gram-Schmidt, which can be computationally expensive. Second, estimating the coefficients via the inner product converges slowly as the basis functions must converge to orthonormality to minimize loss. Below, we provide a faster and more accurate method for computing the coefficients.

**Computing Coefficients via Least Squares.** We propose modifying the function encoder training procedure to compute the coefficients as the solution to a least-squares optimization problem,

$$c := \arg\min_{c \in \mathbb{R}^k} \left\| f - \sum_{j=1}^{k} c_j g_j \right\|_{\mathcal{H}}^2. \tag{6}$$

The least-squares problem in (6) admits a closed-form solution. The coefficients $c$ for function $f$ are computed as

$$c = \begin{bmatrix} \langle g_1, g_1 \rangle_{\mathcal{H}} & \cdots & \langle g_1, g_k \rangle_{\mathcal{H}} \\ \vdots & \ddots & \vdots \\ \langle g_k, g_1 \rangle_{\mathcal{H}} & \cdots & \langle g_k, g_k \rangle_{\mathcal{H}} \end{bmatrix}^{-1} \begin{bmatrix} \langle f, g_1 \rangle_{\mathcal{H}} \\ \vdots \\ \langle f, g_k \rangle_{\mathcal{H}} \end{bmatrix}, \tag{7}$$

where the inner products are estimated using Monte Carlo integration as in (5). To compare the relative performance

---

**Algorithm 1** Function Encoder Training (LS)

---

**given** source datasets $\{D_{S_1}, \ldots, D_{S_n}\}$, learning rate $\alpha$
Initialize basis $\{g_1, \ldots, g_k\}$ with parameters $\theta$
**while** not converged **do**
    **for all** $D_{S_\ell} \in \{D_{S_1}, \ldots, D_{S_n}\}$ **do**

$$c^\ell = \begin{bmatrix} \langle g_1, g_1 \rangle_{\mathcal{H}} & \cdots & \langle g_1, g_k \rangle_{\mathcal{H}} \\ \vdots & \ddots & \vdots \\ \langle g_k, g_1 \rangle_{\mathcal{H}} & \cdots & \langle g_k, g_k \rangle_{\mathcal{H}} \end{bmatrix}^{-1} \begin{bmatrix} \langle f_{S_\ell}, g_1 \rangle_{\mathcal{H}} \\ \vdots \\ \langle f_{S_\ell}, g_k \rangle_{\mathcal{H}} \end{bmatrix}$$

        $\hat{f}_{S_\ell} = \sum_{j=1}^{k} c_j^\ell g_j$
    **end for**
    $L \leftarrow \frac{1}{n} \sum_{\ell=1}^{n} \|f_{S_\ell} - \hat{f}_{S_\ell}\|_{\mathcal{H}}^2$
    $L_{reg} \leftarrow \sum_{i=1}^{k} (\|g_i\|_{\mathcal{H}}^2 - 1)^2$
    $\theta \leftarrow \theta - \alpha \nabla_\theta (L + L_{reg})$
**end while**
**return** $\{g_1, \ldots, g_k\}$

---

of the two coefficient calculation methods, we distinguish the inner product method (IP) using (4) and the least-squares method (LS) using (7).

The solution in (7) provides a theoretically optimal projection onto the learned basis in the least-squares sense. The key benefit of least squares is that it does not require the basis functions to be orthonormal. Instead, we only presume that the basis functions are linearly independent, a mild assumption (see Lamperski, 2022). In practice, this advantage is significant, and least squares performs better in most cases.

Although least squares is conceptually simple, the use of least squares during the training process imposes additional computational challenges. Primarily, least squares requires additional regularization to ensure the basis function magnitudes remain in an acceptable range. See Appendix A.

**Training Algorithm.** To train the basis functions, we assume access to a dataset $D_{S_\ell}$ for each source task $f_{S_\ell}$. Using this dataset, we compute the coefficients $c^\ell$ for each $f_{S_\ell}$ using either IP or LS, depending on the method chosen. This yields an approximation for $f_{S_\ell}$ via (3). The loss is the mean approximation error over source tasks,

$$L = \frac{1}{n} \sum_{\ell=1}^{n} \left\| f_{S_\ell} - \sum_{j=1}^{k} c_j^\ell g_j \right\|_{\mathcal{H}}^2. \tag{8}$$

The basis functions are trained via gradient descent to minimize this loss. See Algorithm 1 for the pseudo-code.

### 3.2. Online Inference

After training, we may approximate a target task $f_T$ given a small dataset $D_{f_T}$. We use the dataset to compute the coefficients, and approximate $f_T$ as a linear combination

of the basis functions. Notably, the coefficient calculations are computationally simple. The inner product approximation is effectively a sample mean, which can be computed quickly in parallel. Furthermore, if using the least-squares method, the Gram matrix in (7) is of size $k \times k$, which we select as a hyperparameter, meaning the matrix inverse can be computed quickly even for large datasets.

### 3.3. Universal Function Space Approximation Theorem

An important question is which properties a Hilbert space must have to be well represented by learned basis functions. In this section, we provide an existence proof to show that a function encoder can represent any separable Hilbert space with arbitrary precision.

**Theorem 1.** *Let $K \subset \mathbb{R}^n$ be compact. Define the inner product $\langle f, g \rangle_{\mathcal{H}} := \int_K f(x)^\top g(x) dx$ and the induced norm $\|f\|_{\mathcal{H}} := \sqrt{\langle f, f \rangle_{\mathcal{H}}}$. Let $\mathcal{H} = \{f : K \to \mathbb{R}^m | f \text{ continuous}, \|f\|_{\mathcal{H}} < \infty\}$ be a separable Hilbert space. Then, there exist neural network basis functions $\{\hat{e}_1, \hat{e}_2, ...\}$ such that for any $\epsilon > 0$ and for any function $f \in \mathcal{H}$, there exists $N \in \mathbb{N}$ and $c \in \mathbb{R}^N$ such that*

$$\|f - \sum_{i=1}^{N} c_i \hat{e}_i\|_{\mathcal{H}} < \epsilon \|f\|_{\mathcal{H}}.$$

*Proof (Sketch).* Every separable Hilbert space $\mathcal{H}$ has a countable orthonormal basis $\{e_1, e_2, \ldots\}$ (Oden & Demkowicz, 2018, Theorem 6.3.1), and any function $f \in \mathcal{H}$ can be represented in terms of the basis. Let $\hat{e}_i$ be a neural network approximation of $e_i$. By the universal approximation theorem of neural networks (Leshno et al., 1993), such a neural network always exists with arbitrary error. Bound the error of each basis function approximation as a decreasing geometric series. Then, the overall error of representing any function as a linear combination of these approximations is finite and arbitrarily small.

The proof shows that in the limit of infinite basis functions, any member of the Hilbert space is well approximated by a function encoder. See Appendix B for the full proof and a discussion.

## 4. Experimental Comparison of Transfer

We compare our proposed approach against existing transfer approaches on challenging transfer scenarios. Specifically, we compare function encoders using the inner product (IP) method (Ingebrand et al., 2024b), function encoders using the least-squares (LS) method (ours), auto encoders, transformers (Vaswani et al., 2017), transformer functional encodings (TFE), an oracle with privileged information, MAML (Finn et al., 2017), and two naive algorithms: brute force (BF) and brute force to basis (BFB). We

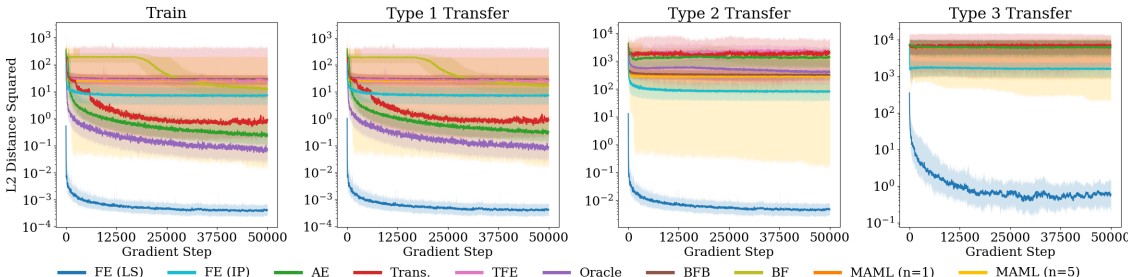

*Figure 2.* **Empirical Results on the Polynomial Dataset.** While many approaches demonstrate moderate type 1 transfer, only the function encoder successfully achieves all three types, as illustrated by its orders of magnitude advantage over other approaches.

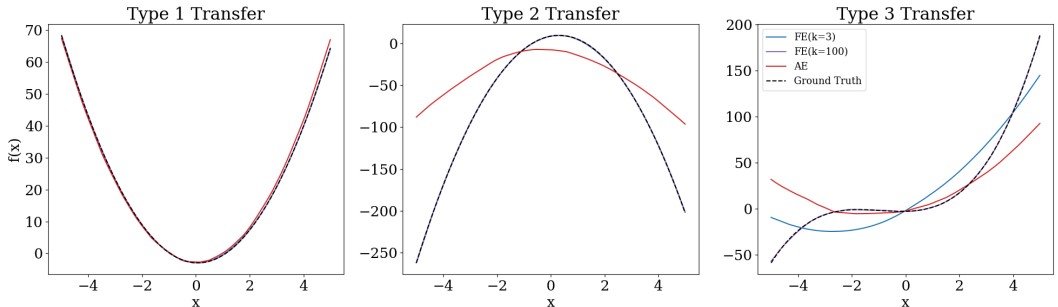

*Figure 3.* **Qualitative Analysis of Transfer on the Polynomial Dataset.** In this illustrative example, we visualize the function encoder and one baseline, the auto encoder, on each of the three types of transfer. We observe that both approaches achieve reasonable performance for type 1 transfer. For type 2 transfer, the target function is much larger in magnitude than any function in the training set. The auto encoder fails at this function because it has only learned to output functions from the training function space. In contrast, the function encoder generalizes to the entire span of the training function space by design. For type 3 transfer, the target function is a cubic function. The auto encoder nonetheless outputs a function that is similar to the ones seen during training. When using a function encoder with only three basis functions, the basis functions only span the three-dimensional space of quadratic functions, and so its approximation is the best quadratic to fit the data. When using 100 basis functions, the basis functions spans the space of quadratics, but additionally have 97 unconstrained dimensions. Due to the use of least squares, the function encoder with 100 basis functions optimally uses these extra 97 dimensions to fit the new function. Therefore, it is able to reasonable approximate this function as well, despite having never seen a cubic function during training.

assess the capability of these existing approaches to transfer to new tasks that have varying geometric relationships to the set of observed source predictors. See Appendix I for more information on the baselines.

We consider four benchmark transfer tasks: 1) a polynomial regression task to illustrate the proposed categories, 2) a CIFAR image classification task, 3) an inference task on the 7-Scenes dataset, and 4) a dynamics estimation task on MuJoCo data. See Appendix H for more information. We show that the proposed approach, FE (LS), transfers to new tasks, even when the tasks are outside the convex hull of source predictors. The algorithm learns the underlying structure and can transfer knowledge to new domains.

### 4.1. An Illustrative Polynomial Regression Task

We first consider transfer on a simple polynomial regression task. This problem is designed to be simple while still demonstrating the types of transfer and failure modes of

current approaches. The models are trained using polynomials $\{ax^2 + bx + c \mid a, b, c \in [-3, 3]\}$. We evaluate type 1 transfer by sampling unseen functions from the same space. We evaluate type 2 transfer be sampling polynomials with coefficients $a, b, c \in [-20, 20]$. We then consider extrapolation to a new function class, $\{f(x) = ax^3 + bx^2 + cx + d \mid a, b, c, d \in [-3, 3]\}$, to evaluate type 3 transfer.

In Figure 2, we see that existing approaches achieve moderate type 1 transfer, *interpolation in the convex hull*. However, existing approaches fail to extrapolate to the linear span (type 2) or to other functions in $\mathcal{H}$ (type 3), while the function encoder using least squares achieves low $L^2$ error.

The function encoder's excellent performance is the result of least squares, which is the optimal projection of the target task onto the learned basis. For type 1 and type 2 transfer, this projection is almost a perfect recreation of the target task, since the basis functions are trained to span the source tasks, and the target task lies in the span of the

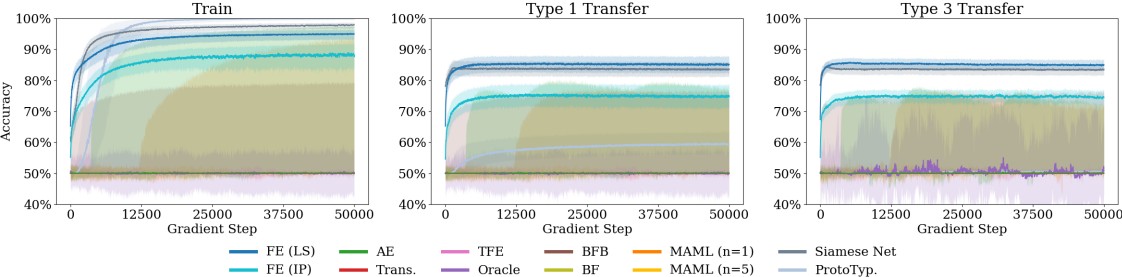

*Figure 4.* **Empirical Results on the CIFAR Dataset.** The training curves show the two ad-hoc baselines seem to be performing best, and many algorithms fail to converge on all or some seeds. However, when measuring type 1 transfer, the function encoder performs best, achieving slightly better performance than Siamese networks. For type 3 transfer, few-shot classification of unseen classes, the function encoder again performs best, albeit similar to Siamese networks. The key idea is that function encoders are performing comparably to ad-hoc approaches despite being designed for a more general setting.

source tasks. For type 3 transfer, this projection necessarily has error, but the error is minimized given the learned basis. See Figure 3 for a visualization of how this property greatly improves transfer.

### 4.2. CIFAR Dataset: Classifying Unseen Objects

We evaluate the function encoder and relevant baselines on the CIFAR 100 dataset (Krizhevsky, 2009). Specifically, we evaluate on the few-shot classification problem, where the classifier is given a set of images belonging to a class and a set of counterexamples that belong to other classes, and must determine if a new image belongs to the specified class. The classifier function corresponding to a given class is one function from the function space. For example, the "apple" classifier returns true if an image contains an apple and false otherwise. 90 classes are used during training, and 10 are held out for testing. Type 1 transfer is evaluating a model's performance on unseen images from the training classes. Indeed, these are the training functions, but for new inputs. Type 2 transfer is not readily testable with this dataset. Conceptually, linear combinations of classes would correspond to images which belong to both classes. An "apple" classifier plus a "green" classifier would correspond to a "green apple" classifier. Type 3 transfer is the model performance on unseen classes. See Figure 19 for a visualization of the data setting.

The function encoder uses a probability distribution-based inner product; See Appendix C.2 for more information. To make a fair comparison, we also train other algorithms, where applicable, using the same inner produced-based distance function. In Appendix H.1, we include an experiment for other algorithms using a cross-entropy loss function. Furthermore, as is common for image based classification problems, weight-regularization is necessary to prevent overfitting for some algorithms, including the function encoder. In addition to the standard baselines, we also use two ad-hoc baselines: Siamese networks (Bromley et al.,

1993) and prototypical networks (Snell et al., 2017).

Our results indicate that the function encoder and the two ad-hoc baselines perform best. While many approaches have some convergent seeds, most do not converge consistently, indicating the difficulty of learning in this setting. Furthermore, the two ad-hoc baselines perform best on the training set, but demonstrate a significant drop in performance in a type 1 or type 3 setting. The best algorithm overall is the function encoder, although Siamese networks are similar. The key result of this experiment is that the function encoder is achieving comparable performance to ad-hoc algorithms designed explicitly for this setting, despite the fact that the function encoder is more general.

### 4.3. 7-Scenes Dataset: Estimating Position from Images

We use the 7-Scenes dataset (Shotton et al., 2013), where the goal is to estimate the position of a camera from an image. The example dataset consists of pairs of images and their locations within the scene. As the name suggests, there are seven unique scenes in this dataset, along with various trajectories of images in each scene. Six scenes are used for training, and one is heldout.

Type 1 transfer is again evaluated on the training scenes, but on heldout images. Type 2 transfer is not readily testable, though it conceptually corresponds to a change in units, e.g., changing the position from meters to centimeters, or changing the origin. Type 3 transfer corresponds to the performance on the unseen scene. See Figure 21 for a visualization of the data setting.

The results indicate that while most algorithms achieve strong training performance, all see significant decreases in performance for type 1 and type 2 transfer. The function encoder however achieves the best transfer due to its use of least squares. Effectively, the function learns a useful set of features for the training dataset, and then optimally leverages these features for transfer, even though these features

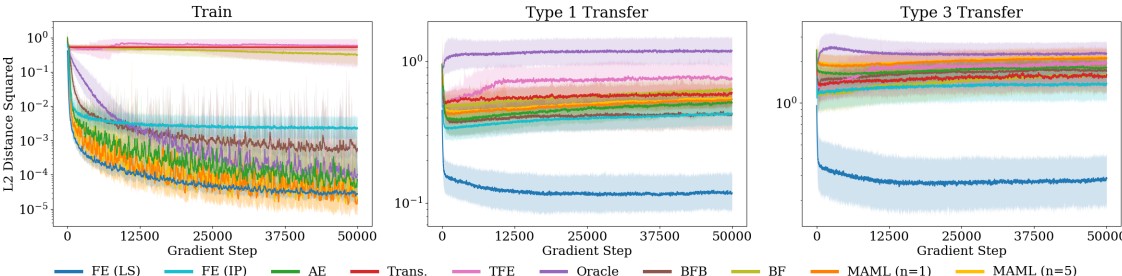

*Figure 5.* **Empirical Results on the 7-Scenes Dataset.** Many approaches converge during training. As expected, all approaches perform much worse at type 1 transfer, indicating a degree of over-fitting. The function encoder performs best at both type 1 and type 3 transfer, indicating its ability to optimally use the learned features for unseen data.

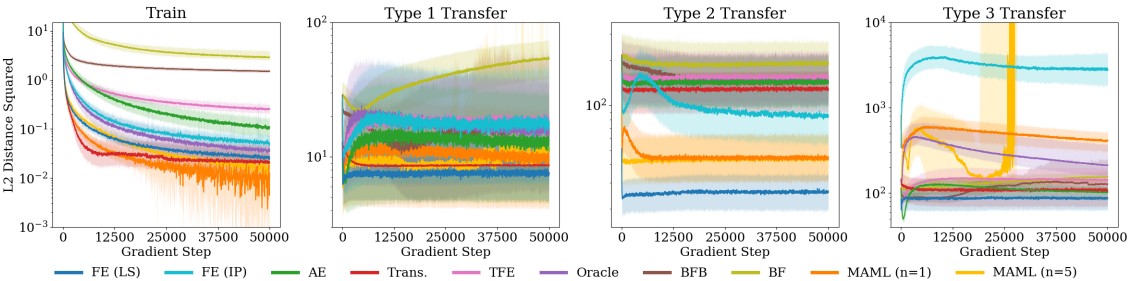

*Figure 6.* **Empirical Results on the Ant Dataset.** All algorithms demonstrate convergence during training, albeit to various levels. However, many algorithms perform much worse for type 1 transfer. The function encoder performs best, although many approaches, such as the transformer and MAML(n=5), are comparable. Furthermore, the function encoder is clearly best for type 2 transfer. Type 3 transfer tells an interesting story. The function encoder has the best stable performance, although approaches such as the transformer and the auto encoder are not far behind. At the beginning of training, the auto encoder shows the best overall performance, although it degrades as training continues. This is because training is not optimizing for type 3 transfer, and the best model parameters for the training dataset are *not* the best model parameters for type 3 transfer. Thus, its performance is unstable.

are likely suboptimal for the unseen images.

### 4.4. MuJoCo Dataset: Estimating Unseen Dynamics

We adapt the hidden-parameter MuJoCo Ant dataset from Ingebrand et al. (2024a). This dataset consists of a MuJoCo Ant (four-legged robot) walking on a flat surface. The lengths of the robot's limbs, and the control authority, are varied which effectively changes the dynamics function. The goal is to predict the next state of the robot given the current state and action, where the example dataset provides data from the current set of hidden parameters.

During training, hidden parameters are sampled from 0.5x to 1x the default robot values. Type 1 transfer consists of unseen hidden-parameters sampled from that same space. Type 2 transfer consists of synthetically generated dynamics which correspond to linear combinations of the dynamics sampled from the type 1 parameter space. Type 3 transfer consists of hidden parameters sampled from 1.5x to 2x the default values, which corresponds to a much larger robot. See Appendix H.3 for more details on this dataset. See Figure 6 for experimental results.

The results indicate that many approaches achieve some but not all types of transfer on this dataset. Meta learning

achieves good type 1 transfer, but fails at type 2 and type 3. Interestingly, the auto encoder briefly achieves great type 3 transfer, but its performance degrades as training progresses. This is because minimizing training loss may directly conflict with minimizing type 3 transfer, and type 3 transfer by definition cannot be used for training. The function encoder achieves the overall best transfer, although some algorithms may be comparable for a given type of transfer. Additionally, for simplicity the function encoder did not use either the residuals method (see Appendix E) or neural ODE basis functions, which are minor changes that have been shown to lead to great improvements in accuracy for this problem (Ingebrand et al., 2024b;a).

### 5. Conclusion

We have introduced a novel categorization of transfer learning based on the geometric interpretation of Hilbert spaces. We made novel improvements to the function encoder algorithm, and argued that it is a natural solution to transfer learning in Hilbert spaces. We empirically validated this argument using four transfer learning datasets.

We leave many open questions for future work. Primarily, applying the function encoder algorithm to other set-

tings requires a well-defined inner product. Therefore, designing inner products for sequences and graphs is a necessary and interesting challenge. Furthermore, this work does not address how to efficiently represent a low-dimensional manifold lying in an infinite dimensional space. In such a setting, naively learning basis functions may require an unpractical number of basis functions to reasonable span the manifold, while the manifold itself is low-dimensional. Thus there may be similar algorithms which learn a basis only for the manifold.

## Impact Statement

This paper presents work whose goal is to advance the field of Machine Learning. There are many potential societal consequences of our work, none which we feel must be specifically highlighted here.

## Acknowledgments

This work is supported by DARPA HR0011-24-9-0431, NSF 2214939, RTX CW2231110, and AFOSR FA9550-19-1-0005. Any opinions, findings, conclusions or recommendations expressed in this material are those of the authors and do not necessarily reflect the views of the funding organizations.

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

## A. Basis Function Regularization

As with all (unregularized) least-squares approaches, the Gram matrix can occasionally be close to singular which makes the matrix inverse numerically unstable during training. Thus, we add a small regularization $\lambda$ to the diagonal of the Gram matrix in the style of Ridge regression, which improves the numerical stability of the inverse. Yet, the regularization term causes the eigenvalues to increase, meaning the solution to the regularized least-squares problem will be a slight under-approximation. Since the basis functions are trained to minimize the error of the approximation, and the approximation is an underestimate, the basis functions constantly grow in magnitude. Thus, we additionally impose an overall regularizer on the norm of the basis functions to be close to 1 to prevent this:

$$L_{reg} = \sum_{i=1}^{k} (\|g_i\|_{\mathcal{H}}^2 - 1)^2 \tag{9}$$

Note that we do not need the basis functions to be unit, we only require that they do not grow infinitely in magnitude as training progresses.

## B. Universal Function Space Approximation Theorem

**Theorem 1** (Restated). *Let $K \subset \mathbb{R}^n$ be compact. Define the inner product $\langle f, g \rangle_{\mathcal{H}} := \int_K f(x)^\top g(x) dx$ and the induced norm $\|f\|_{\mathcal{H}} := \sqrt{\langle f, f \rangle_{\mathcal{H}}}$. Let $\mathcal{H} = \{ f : K \to \mathbb{R}^m \mid f \text{ continuous}, \|f\|_{\mathcal{H}} < \infty \}$ be a separable Hilbert space. Then, there exists neural network basis functions $\{\hat{e}_1, \hat{e}_2, ...\}$ such that for any $\epsilon > 0$ and for any function $f \in \mathcal{H}$, there exists $N \in \mathbb{N}$ and $c \in \mathbb{R}^N$ such that*

$$\|f - \sum_{i=1}^{N} c_i \hat{e}_i\|_{\mathcal{H}} < \epsilon \|f\|_{\mathcal{H}}.$$

This theorem states that any separable Hilbert space can be arbitrarily well approximated with (a potentially infinite number of) sufficiently large neural network basis functions.

We will build on the universal function approximation theorem for neural networks (Hornik et al., 1989; Cybenko, 1989). We use the results stated in (Leshno et al., 1993):

**Theorem 2** (Leshno et al., 1993). *Let $\sigma$ be a non-polynomial activation function such that the closure of its discontinuous points is of zero Lebesgue measure. Then for $n \in \mathbb{N}$, compact $K \subset \mathbb{R}^n$, $f \in C(K, \mathbb{R})$, $\delta > 0$, there exists neural network parameters $\ell \in \mathbb{N}$, $A \in \mathbb{R}^{\ell \times n}$, $B \in \mathbb{R}^\ell$, $D \in \mathbb{R}^\ell$ such that*

$$\sup_{x \in K} |f(x) - \hat{f}(x)| < \delta,$$

*where $\hat{f}(x) = D \cdot (\sigma(A \cdot x + B))$.*

This theorem states that any continuous, scalar-valued function can be arbitrarily well approximated by a sufficiently wide neural network.

*Proof.* First, we will extend the universal function approximation theorem from continuous, scalar-valued functions to continuous, vector-valued functions. Suppose the functions map to $\mathbb{R}^m$ instead of $\mathbb{R}$, and consider the Euclidean norm. Then there exists a neural network such that $\|f(x) - \hat{f}(x)\|_2^2 = (f(x) - \hat{f}(x))^\top (f(x) - \hat{f}(x)) < \delta^2 m$ for all $x$, where the only change to the neural network is that $D \in \mathbb{R}^{\ell \times m}$. This results from the fact that we can treat each output dimension independently, and each output dimension has arbitrarily small error. Thus the sum of the squared errors is also arbitrarily small.

Since the point-wise error is bounded in a Euclidean sense and $K$ has finite measure, then $\|f - \hat{f}\|_{\mathcal{H}} = \int_K (f(x) - \hat{f}(x))^\top (f(x) - \hat{f}(x)) dx < \int_K \delta^2 m \, dx = V \delta^2 m$, where $V$ is the volume of $K$. Therefore, the universal approximation theorem for neural networks implies that any function $f \in \mathcal{H}$ can be arbitrarily well approximated by a sufficiently wide neural network.

Next, we will leverage the fact that every separable Hilbert space has a countable orthonormal basis (Oden & Demkowicz, 2018, Theorem 6.3.1). Let $\{e_1, e_2, ...\}$ be such a basis. Then for any $f \in \mathcal{H}$,

$$f = \sum_{i=1}^{\infty} c_i e_i := \lim_{N \to \infty} \sum_{i=1}^{N} c_i e_i,$$

where $c_i = \langle f, e_i \rangle$. This sequence converges because $\text{span}\{e_1, e_2, ...\}$ is dense in $\mathcal{H}$ (Oden & Demkowicz, 2018, Chapter 6.3). Also, these basis functions are members of $\mathcal{H}$, and are thus continuous.

Let $\hat{e}_i$ be the neural network approximation of $e_i$ with $||\hat{e}_i - e_i||_{\mathcal{H}} < \frac{\delta}{2^i}$, for some $\delta > 0$. By the universal function approximation theorem (extended to the vector-valued case), such a neural network always exists.

Each $\hat{e}_i$ can be decomposed into $e_i$ and an error vector $d_i$,

$$\hat{e}_i = e_i + d_i.$$

By definition,

$$||d_i||_{\mathcal{H}} < \frac{\delta}{2^i}.$$

Let

$$\hat{f}_N = \sum_{i=1}^{N} c_i \hat{e}_i = \sum_{i=1}^{N} c_i (e_i + d_i) = \sum_{i=1}^{N} c_i e_i + \sum_{i=1}^{N} c_i d_i.$$

Consider $\lim_{N \to \infty} ||\hat{f}_N - f||_{\mathcal{H}}$.

$$\lim_{N \to \infty} ||\hat{f}_N - f||_{\mathcal{H}} = \lim_{N \to \infty} ||\sum_{i=1}^{N} c_i e_i + \sum_{i=1}^{N} c_i d_i - f||_{\mathcal{H}}$$

$$\leq \lim_{N \to \infty} ||\sum_{i=1}^{N} c_i e_i - f||_{\mathcal{H}} + ||\sum_{i=1}^{N} c_i d_i||_{\mathcal{H}}$$

Notice that $||\sum_{i=1}^{N} c_i d_i||_{\mathcal{H}} \leq \sum_{i=1}^{N} |c_i| \, ||d_i||_{\mathcal{H}}$ by the triangle inequality. Also, by the definition of $d_i$, $\sum_{i=1}^{N} |c_i| \, ||d_i||_{\mathcal{H}} < \sum_{i=1}^{N} |c_i| \frac{\delta}{2^i}$.

Furthermore, for a given $f$ of size $||f||_{\mathcal{H}}$, the largest value of $|c_i|$ is achieved if $f = a \cdot e_i$ for some $a \in \mathbb{R}$, $i \in \mathbb{N}$, and in this case $|c_i| = |a| = ||f||_{\mathcal{H}}$. Thus, we can bound $|c_i| \leq ||f||_{\mathcal{H}}$. Therefore,

$$||\sum_{i=1}^{N} c_i d_i||_{\mathcal{H}} < \delta ||f||_{\mathcal{H}} \sum_{i=1}^{N} \frac{1}{2^i}.$$

We can rewrite the limit above as

$$\lim_{N \to \infty} ||\hat{f}_N - f||_{\mathcal{H}} < \lim_{N \to \infty} ||\sum_{i=1}^{N} c_i e_i - f||_2 + \delta ||f||_{\mathcal{H}} \sum_{i=1}^{N} \frac{1}{2^i}.$$

$\sum_{i=1}^{N} \frac{1}{2^i}$ is a geometric series whose limit is 1 as $N \to \infty$. Since both components are finite in the limit,

$$\lim_{N \to \infty} ||\hat{f}_N - f||_{\mathcal{H}} < \lim_{N \to \infty} \left\|\sum_{i=1}^{N} c_i e_i - f\right\|_{\mathcal{H}} + \lim_{N \to \infty} \delta ||f||_{\mathcal{H}} \sum_{i=1}^{N} \frac{1}{2^i} = \delta ||f||_{\mathcal{H}}.$$

Thus, in the limit of infinite basis functions, the error of the approximation is proportional to the magnitude of the function and an arbitrarily small constant $\delta$. To achieve an error of $||\hat{f}_N - f||_{\mathcal{H}} < \epsilon ||f||_{\mathcal{H}}$ with a finite number of basis functions

$N$, we only need to choose a $\delta$ such that $\epsilon > \delta > 0$. Then we can find a sufficiently large $N$ such that $||\hat{f}_N - f||_{\mathcal{H}} < \delta + (\epsilon - \delta) = \epsilon$. This concludes the proof,

$$\forall \epsilon > 0, f \in \mathcal{H}, \exists N \in \mathbb{N}, c \in \mathbb{R}^N :$$

$$||f - \sum_{i=1}^{N} c_i \hat{e}_i||_{\mathcal{H}} < \epsilon ||f||_{\mathcal{H}}.$$

$\square$

**Discussion.** In the following, we discuss several implications of the proof. First, this proof *does not* imply that an infinite dimensional function space only needs a finite dimensional basis. A infinite dimensional function space unavoidably requires infinite basis functions. However, for any given function in this space, we only need a finite number of basis functions to achieve an arbitrary error. Second, this proof implies that for a finite dimensional space, we only need a finite number of learned basis functions and we can achieve an arbitrarily small error for all functions in this function space. Lastly, the requirement that each basis function $\hat{e}_i$ has smaller error than the basis function $\hat{e}_{i-1}$ implies that the size of basis function $\hat{e}_i$ will tend to be bigger than the basis function $\hat{e}_{i-1}$. In other words, increasing basis function precision likely increases neural network width.

## C. Choosing an Inner Product

The key design decision for function encoders is the choice of an appropriate and well-defined inner product. Fortunately, many prior works have determined viable inner products for many spaces of interest, such as deterministic function spaces and probability distributions. In general, we define a valid inner product over the output space $\mathcal{Y}$, where $\mathcal{Y}$ is a Hilbert space also, and use the Bochner-Lebesgue integral to define the inner product for $\mathcal{H} := \{f : \mathcal{X} \to \mathcal{Y}\}$,

$$\langle f, g \rangle_{\mathcal{H}} := \int_{\mathcal{X}} \langle f(x), g(x) \rangle_{\mathcal{Y}} dx. \tag{10}$$

Vector addition and scalar multiplication for $\mathcal{H}$ are defined pointwise:

$$(f + g)(x) := f(x) + g(x) \tag{11}$$

$$(\alpha f)(x) := \alpha f(x) \tag{12}$$

Thus, function encoders can be extended to many function spaces with minimal modifications. Below, we describe three common inner products, along with some equivalent forms that improve computational efficiency.

### C.1. Euclidean Vectors

The most common output space is a real vector space $\mathcal{Y} = \mathbb{R}^m$. We use the standard Euclidean vector operations. The resulting inner product for $\mathcal{H} := \{f : \mathcal{X} \to \mathbb{R}^m\}$ is

$$\langle f, g \rangle_{\mathcal{H}} := \int_{\mathcal{X}} f(x)^\top g(x) dx. \tag{13}$$

### C.2. Discrete Probability Distributions

Another common output space is discrete probability distributions over classes. The distributions themselves are unobserved but we instead observe samples from the distributions. We leverage the Hilbert space definitions for discrete probability distributions from Egozcue et al. (2003). The $D$-class probability distribution is defined as

$$\mathcal{S}^D := \left\{ x = [x_1, x_2, ..., x_D] : x_i > 0, \sum_{i=1}^{D} x_i = 1 \right\}. \tag{14}$$

Note that this implies the probability of any class is always greater than 0. This is necessary to ensure the space is a Hilbert space. The corresponding vector operations are defined as follows:

$$x + y := \left[ \frac{x_1 y_1}{\sum_{i=1}^{D} x_i y_i}, \frac{x_2 y_2}{\sum_{i=1}^{D} x_i y_i}, ...., \frac{x_D y_D}{\sum_{i=1}^{D} x_i y_i} \right] \tag{15}$$

$$\alpha x := \left[ \frac{x_1^\alpha}{\sum_{i=1}^{D} x_i^\alpha}, \frac{x_2^\alpha}{\sum_{i=1}^{D} x_i^\alpha}, ...., \frac{x_D^\alpha}{\sum_{i=1}^{D} x_i^\alpha} \right] \tag{16}$$

where $x$ and $y$ are in $\mathcal{S}^D$. The inner product is defined as

$$\langle x, y \rangle_{\mathcal{S}^D} := \sum_{i=1}^{D} \log \frac{x_i}{G(x)} \log \frac{y_i}{G(y)}, \tag{17}$$

where $G(x) = (x_1 x_2 ... x_D)^{1/D}$ is the geometric mean. Under these definitions, $\mathcal{S}^D$ is a Hilbert space (Egozcue et al., 2003). However, as vector addition requires multiplication and scalar multiplication requires exponentiation, these definitions are computationally inefficient.

Instead, we can leverage equivalent definitions that use addition and multiplication instead of multiplication and exponentiation, respectively. To do so, we will define equivalent operations in logit space $l^D$, which is simply $\mathbb{R}^D$ with a modified inner product. We define $logit : \mathcal{S}^D \to l^D$ as

$$logit(x) = [\log(x_1), ..., \log(x_D)] \tag{18}$$

and $probability : l^D \to \mathcal{S}^D$ as

$$probability(y) = \left[ \frac{e^{y_1}}{\sum_{i=1}^{D} e^{y_i}}, ...., \frac{e^{y_D}}{\sum_{i=1}^{D} e^{y_i}} \right] \tag{19}$$

These two operations allow us to move from probability space to logit space and vice versa. Note that logit space is $D$-dimensional while probability space is actually $(D-1)$-dimensional. Effectively, we are working with a quotient space where members of $\mathbb{R}^D$ are equivalent if they map to the same probability distribution. Logit space is convenient because the vector operations become the standard operations in $\mathbb{R}^D$, i.e. they use addition for vector addition and multiplication for scalar multiplication. These operations are significantly cheaper and produce more stable gradients than the equivalent probability space operations. The inner product for logit space $l^D$ is

$$\langle x, y \rangle_{l^D} := \sum_{i=1}^{D} (x_i - \mu(x))(y_i - \mu(y)), \tag{20}$$

where $\mu(x) = \frac{1}{D} \sum_{i=1}^{D} x_i$ is the mean of $x$. Under these definitions, we can work entirely in logit space and only convert to probability distributions when necessary.

**Practical Data Considerations** In practice, we do not observe the probability distribution directly. Instead, we observe one sample point from the distribution. Therefore, we take the maximum likelihood perspective and would assign all probability mass to the observed class and 0 to the others. However, this would violate the requirement that the probability of a given class is non-zero. Therefore, we assign a high probability (or a positive logit) to the observed class and low probability (or negative logits) to the other classes. The exact value chosen for the high probability (or logit) is a hyper-parameter, and affects the model's confidence. For example, choosing $80\%$ probability for the observed class will make the model appear uncertain, while choosing $99.9\%$ probability will make the model outputs appear very certain. Thus, this parameter should be carefully chosen.

**Conditional Discrete Probability Distributions** Typically, we are concerned with probability distributions conditioned on some input, i.e. assigning a class to an input. In this case, we simply use the Bochner-Lebesgue integral to define the inner product for functions mapping to discrete probability distributions. Thus, the function encoder is applicable to conditional, discrete probability distributions that are prevalent in classification problems. Below, we illustrate what this looks like for the CIFAR dataset.

Let the input space $\mathcal{X}$ be the set of possible images and the output space $\mathcal{Y}$ consists of the categories `True` and `False`. The classifier for a given object, $f_{obj}$, determines if an image contains the object. In other words, if the specified object appears

in the image $x$, $f_{obj}(x) =$ `True`. If not, $f_{obj}(x) =$ `False`. A dataset $D = \{(x_i, y_i)\}_{i=1}^m$ consists of pairs of images and `True`/`False` labels. Let $l_{pos}$ and $l_{neg}$ be the positive and negative logits as described above. Then `True` $\approx [l_{pos}, l_{neg}]$ and `False` $\approx [l_{neg}, l_{pos}]$. In other words, the first dimension of the distribution corresponds to the (log) probability that the object is contained in the image, and the second dimension corresponds to the (log) probability that object is not contained in the image. Thus, the `True`/`False` labels are converted to distributional representations. Lastly, we only need to define the approximate inner product for two classifiers $f$ and $h$ given a dataset of $m$ images and labels:

$$\langle f, h \rangle = \int_{\mathcal{X}} \langle f(x), h(x) \rangle_{l^D} \, dx \approx \frac{1}{m} \sum_{i=1}^m \langle f(x_i), h(x_i) \rangle_{l^D}$$

In summary, we have defined an inner product for conditional distributions, and can thus directly apply the function encoder algorithm to model conditional distributions.

## C.3. Continuous Probability Distributions

Similar to discrete distributions, it is often useful to estimate probability densities in continuous spaces. We leverage the definitions of Egozcue et al. (2006), which is analogous to the discrete distribution case with integration in place of summation. For a finite interval $(a, b)$,

$$A(a, b) := \{f : (a, b) \to \mathbb{R}, f > 0 \text{ a.e.}, \log f \in L^2(a, b)\} \tag{21}$$

The vector operations are

$$(f + h)(x) := \frac{f(x)h(x)}{\int_a^b f(\sigma)h(\sigma)d\sigma}, \tag{22}$$

$$(\alpha f)(x) := \frac{f(x)^\alpha}{\int_a^b f(\sigma)^\alpha d\sigma}. \tag{23}$$

Note the similarity to the discrete case. Lastly, the inner product is

$$\langle f, h \rangle_{A(a,b)} := \int_a^b \log \frac{f(x)}{G(f)} \log \frac{h(x)}{G(h)} dx, \tag{24}$$

where $G(f) = \exp(\frac{1}{V} \int_a^b \log(f(x)))$ is the geometric mean of the function $f$. Thus, we can learn probability distributions using a function encoder. Similar to the discrete case, we can improve computational efficiency by using logits. Thus we define an equivalent logit space $l(a, b) := \{f : (a, b) \to \mathbb{R}, f \in L^2(a, b)\}$. Vector addition and scalar multiplication for logit space are the traditional point-wise operations. The inner product is defined as

$$\langle f, h \rangle_{l(a,b)} := \int_a^b (f(x) - \mu(f))(h(x) - \mu(h))dx, \tag{25}$$

where again $\mu(f) := \frac{1}{V} \int_a^b f(x)dx$ is the mean of $f$.

The continuous distribution case is mostly analogous to the discrete case, but with a few important distinctions. Given $d$ samples from some distribution $P(Y = y)$ defined over a set $\mathcal{Y}$, we again wish to use the maximum likelihood perspective. However, while for the discrete case we can iterate all possible classes which were not sampled, in the continuous case there are infinite values in $\mathcal{Y}$ not sampled. Therefore, we must also sample "negative" values from the support of the distribution. In practice, this means either defining a uniform sampling function over $\mathcal{Y}$, and using these uniformly sampled values as negative examples, or using a dataset of values in $\mathcal{Y}$ and sampling uniformly from this dataset. The former case makes sense in simpler problems where we can properly define $\mathcal{Y}$, for example if $\mathcal{Y}$ is an interval $(0, 1)$. However, there are many cases where the set $\mathcal{Y}$ is hard to define, especially in real-world datasets where it is not clear what range of values $\mathcal{Y}$ can take. In this case, we can uniformly sample from a dataset.

These negative samples are used to define points which are not likely, since they were not sampled. Without them, the empirical distribution is uniform over the positive samples, and therefore the distribution of best fit would be uniform over all of $\mathcal{Y}$. The negative samples provide additional information on which samples of $\mathcal{Y}$ are unlikely. Intuitively, the model

only has access to "positive" examples, which means the simplest solution is a uniform model over all of $\mathcal{X}$. The negative samples prevent this behavior.

Analogous to the discrete case, we again assign some large probability mass (positive logits) to the sampled values of $\mathcal{Y}$, and some small probability mass (negative logits) to the negative samples.

Lastly, we again need a method to convert from probability densities to logits and vice versa. Analogous to the discrete case, we define $logit : A(a, b) \to l(a, b)$ as

$$logit(f)(y) = \log(f(y)) \tag{26}$$

and $probability : l(a, b) \to A(a, b)$ as

$$probability(h)(y) = \frac{e^{h(y)}}{\int_a^b e^{h(\sigma)} d\sigma}. \tag{27}$$

## D. The Approximate Inner Product and the Distribution of Inputs

The approximate inner product is an important aspect of the function encoder, and Monte-Carlo integration makes many implicit assumptions. For the sake of simplicity, the following will use the $L^2$ inner product, but holds for a more general Bochner-Lebesgue setting too.

Consider the following Mote Carlo approximation,

$$\langle f, g \rangle := \int_{\mathcal{X}} f(x) g(x) dx \tag{28}$$

$$\approx \frac{V}{m} \sum_{i=1}^{m} f(x_i) g(x_i) \tag{29}$$

This approximation assumes that the distribution over $\mathcal{X}$ is uniform. If this is not the case, then this approximation will not hold. Suppose the inputs are sampled according to a distribution $p$. Then the correct Monte-Carlo approximation with importance sampling is

$$\langle f, g \rangle \approx \frac{1}{m} \sum_{i=1}^{m} \frac{f(x_i) g(x_i)}{p(x_i)}. \tag{30}$$

Unfortunately, importance sampling is not possible in many settings because we cannot estimate the distribution $p$. For example, in an image classification setting, we cannot estimate the probability of seeing a given image. In reinforcement learning, the distribution of states is extremely difficult to estimate, and furthermore varies depending on the policy.

Instead, we consider the *weighted* inner product

$$\langle f, g \rangle_p := \int_{\mathcal{X}} f(x) g(x) p(x) dx. \tag{31}$$

This inner product corresponds specifically to the distribution $p$. If we use importance sampling, the approximation is

$$\langle f, g \rangle_p \approx \frac{1}{m} \sum_{i=1}^{m} \frac{f(x_i) g(x_i) p(x_i)}{p(x_i)}, \tag{32}$$

$$\approx \frac{1}{m} \sum_{i=1}^{m} f(x_i) g(x_i). \tag{33}$$

Thus, by weighting the inner product by the probability distribution, we no longer need to estimate the distribution directly. Put another way, by using (33), we are implicitly redefining the inner to be weighted by the probability distribution. The inner product in (31) effectively assigns more weight to the input values that are more likely to be sampled. This assumption is often reasonable. For example, consider an image classification task. The majority of images appear like white noise, and are not of interest. These images should not affect the inner product calculation for two functions defined on images. The naive Monte Carlo integration automatically reduces the importance of these images in the implicit weighted integral.

We may also have the case that the distribution of inputs and the function change simultaneously. For example, this occurs in reinforcement learning where the distribution of states is heavily dependent on the hidden parameters. In this case, we still cannot estimate $p$, and so we must use the approximation in (33). However, by doing so we now have a separate inner product defined for each function in the training set. This is true even in the extreme case where the distributions are disjoint, i.e., domain transfer.

Interestingly, even in the presence of distribution shifts, if there exists a set of basis functions which spans all of the training functions, these basis functions are still optimal. Regardless of the inner product, these basis functions would still be able to reproduce the training functions, and therefore minimize the loss. However, if this is not the case, for example if the dimensionality of the space is larger than the number of basis functions, this function-specific inner product is going to affect which solutions are locally optimal. Future work should explore what implications Monte Carlo integration has on performance, and what changes should be made to account for it, if any.

## E. The Residuals Method

The residuals method for the function encoder was introduced in Ingebrand et al. (2024b). For the sake of completeness, we include a small discussion below.

In the residuals method, a neural network called the average function is trained to fit the function space. As the average function is a single function, it cannot accurately fit the entire function space; Instead, the function it learns is the geometric center of the training functions. Then, the basis functions are trained to minimize the residual error of the average function. See Figure 7. In effect, this allows the function encoder to learn an affine subspace of the function space which may not include the zero vector.

Formally, the average function $\bar{f}_\theta$ is solving the following minimization problem:

$$\bar{f}_\theta = \arg\min_\theta \frac{1}{n} \sum_{i=1}^n ||f_{S_i} - \bar{f}_\theta||_{\mathcal{H}}^2. \tag{34}$$

This optimization problem is solved via gradient descent on the neural network parameters $\theta$. Then, the basis functions are trained on the residuals. For a given function $f_\ell$, the coefficients are calculated as

$$c^\ell = \arg\min_{c \in \mathbb{R}^k} \left\| (f_\ell - \bar{f}_\theta) - \sum_{j=1}^k c_j^\ell g_j \right\|_{\mathcal{H}}^2. \tag{35}$$

Lastly, the error of the approximation is evaluated as

$$L = \frac{1}{n} \sum_{\ell=1}^n \left\| f_{S_\ell} - \left( \bar{f}_\theta + \sum_{j=1}^k c_j^\ell g_j \right) \right\|_{\mathcal{H}}^2. \tag{36}$$

Note the loss in (36) is only used to train the basis functions.

As a byproduct, using the residuals method means the learned function space may not include the zero vector. However, this may sometimes be advantageous. For example, prior work (Ingebrand et al., 2024b) has demonstrated that the residuals method is effective for dynamics prediction in hidden-parameter Markov decision processes. This is because the zero-function, i.e. dynamics that return 0 for all transitions, is not typically a valid transition function under any set of hidden parameters.

Another benefit is the inductive bias of the function estimate. In online settings, it is often convenient to estimate a function from an insufficient amount of data. Using the residuals method is beneficial because its predictions are already centered around the average function in the training dataset. For example, consider the problem of estimating the transition dynamics of a robot from online data. Initially, there is no example data to compute the coefficients. For the residuals method, setting the coefficients to 0 will still give a decent estimate of dynamics, since it will be the average function. Furthermore, one could constrain the coefficients to be close to 0 through additional regularization, which would ensure that the function estimate does not deviate too far from the training dataset.

*Figure 7.* **An Illustration of the Residuals Method.** The standard function encoder algorithm learns a subspace which most accurately spans the training functions. This necessarily implies that the zero vector is representable using coefficients $c = 0$. In contrast, when using the residuals method, the average function first shifts the origin of the basis functions to the center of the training data. Effectively, this allows function encoder to instead learn an affine subspace, or a linear manifold, of the Hilbert space. When using the residuals method, the zero vector is not necessarily representable, and $c = 0$ corresponds to the average function in the training set.

## F. Architecture

The function encoder algorithm is agnostic to the architecture of the basis functions, so long as they are differentiable. Prior work has shown that using neural ODEs for the basis functions greatly improves the performance of the function encoder on continuous-time dynamics prediction tasks (Ingebrand et al., 2024a). This paper demonstrated that the function encoder inherits the inductive biases of the underlying architecture. We conjecture that this trend will hold for other architectures, such as RNNs, GNNs, transformers, etc.

Another consideration is whether to implement the basis functions as completely separate neural networks run in parallel, or one neural network with multiple output heads. From a computational standpoint, current software is optimized for single neural networks, rather than multiple parallel neural networks, and so one network with multiple output heads is much faster during both training and inference. However, this raises a question of whether one approach performs better than the other. We run ablations to compare the performance of the two approaches. See Figure 8. Note that the total number of parameters is held approximately constant, and so the parallel neural networks are much smaller individually than the single, multi-headed neural network.

The results indicate that the performance is roughly equal for type 1 and type 2 transfer. That is, both methods accurately learn the function space present in the training set. However, parallel basis functions are significantly worse at type 3 transfer. We posit that the reason is basis function diversity. For a multi-headed neural network, a change to one basis function changes all, since the weights are shared. This constantly perturbs every basis function, so a given basis function is unlikely to learn a simple function, or even to remain stagnant. Consequently, the basis functions are diverse. This diversity improves type 3 transfer because diverse basis functions are less likely to be linearly dependent.

In contrast, parallel neural networks allow individual basis function updates. It is therefore possible for the learning procedure to train only a few basis functions, while the rest remain unused, if the dimensionality of the space is low. We validate this intuition empirically. See Figure 9 for a visualization of this problem.

While this phenomenon degrades type 3 transfer, it has an interesting side effect that the parallel strategy trains an approximately minimal number of basis functions. It would therefore be possible to prune the unused basis functions to get a minimal set. This method could be useful in computationally limited settings, where the total number of basis functions and parameters could greatly be reduced after training.

## G. Ablations

**Number of Basis Functions.** We vary the number of basis functions and run FE (LS) on all four datasets. We would expect that using more basis functions improves performance, but there would be diminishing returns. Once the number of basis functions exceeds the dimensionality of the space, performance should stop improving. The experimental results

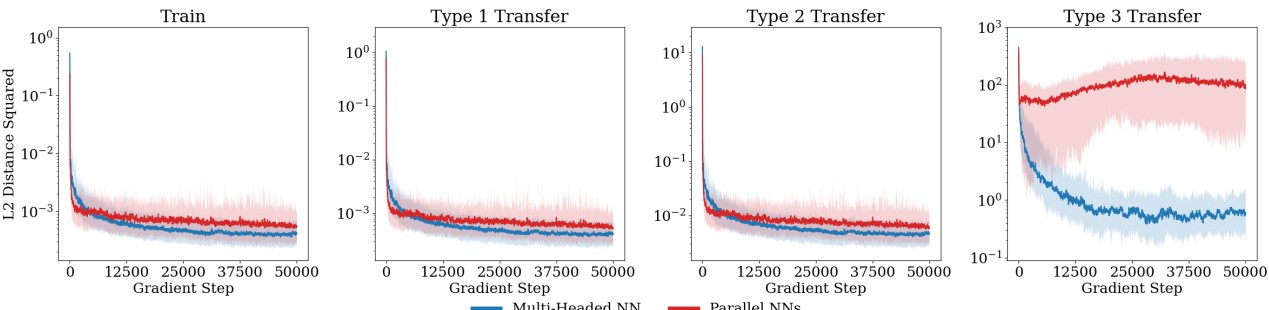

*Figure 8.* **Multi-Headed Neural Network vs Parallel Neural Networks on the Polynomial Dataset.** For type 1 and type 2 transfer, both architecture methods perform roughly equally. However, parallel neural networks perform much worse at type 3 transfer.

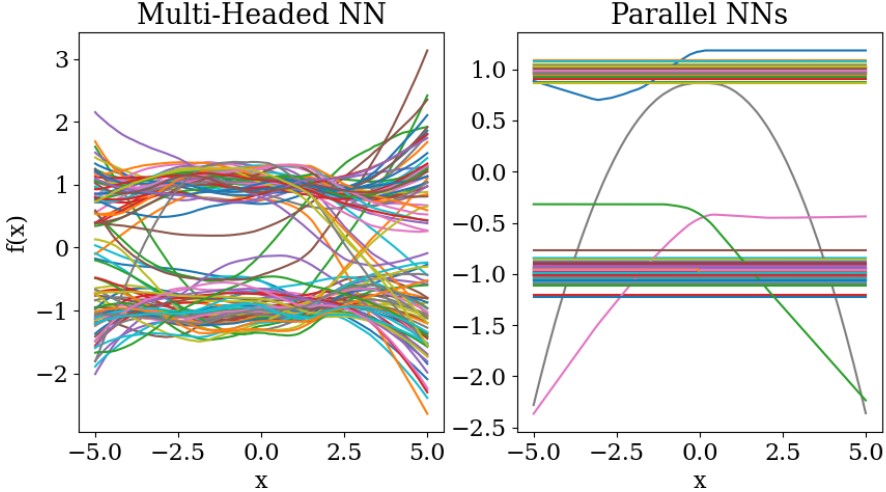

*Figure 9.* **A Comparison of Learned Basis Functions on the Polynomial Dataset.** Each line indicates one basis function. Left: One multi-headed neural network learns a diverse set of basis functions. Right: Parallel neural networks learn a small set of diverse basis functions, while the majority lack diversity or interesting structure.

agree with this hypothesis. See Figures 11, 12, 13, and 14. The figures show the performance at the end of training, averaged over the last 50 evaluations. 3 seeds are run per quantity of basis functions, and figures show min, median, and max values over the seeds. We also consider the cost of increasing the number of basis functions with respect to the training time, and find that it is minor. See Figure 10. Therefore, a reasonable method to choose $k$ is to simply overestimate the dimensionality of the space. We find that 100 basis functions are enough for many problems.

**Number of example data points**   We vary the number of example data points and evaluate the effect on performance. We would expect that using more example data improves performance, but with diminishing returns. The experimental results, shown in Figures 15, 16, 17, and 18, agree with these claims. The results suggest that a minimal amount of data is required for the function encoder to make accurate predictions, and that the minimal amount depends on the dataset. Similar to the above ablation, the figures show 3 seeds evaluated after training, with their performance averaged over the last 50 evaluation steps.

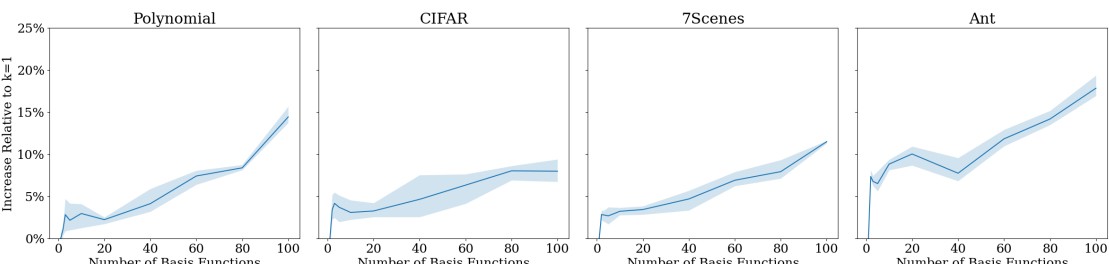

*Figure 10.* **An Ablation on the Number of Basis Functions vs Compute Time.** This figures shows the percentage increase in training time relative to one basis function. The figure shows that the increase in cost is less than 20% for up to 100 basis functions.

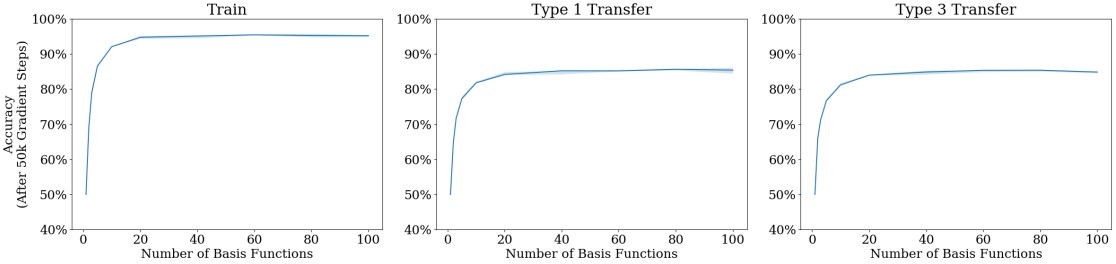

*Figure 11.* **An Ablation on the Number of Basis Functions - Polynomial Dataset.** This figure shows the final performance of FE (LS) after training for a varying number of basis functions. The performance initially improves as the number of basis functions increases, and then levels off.

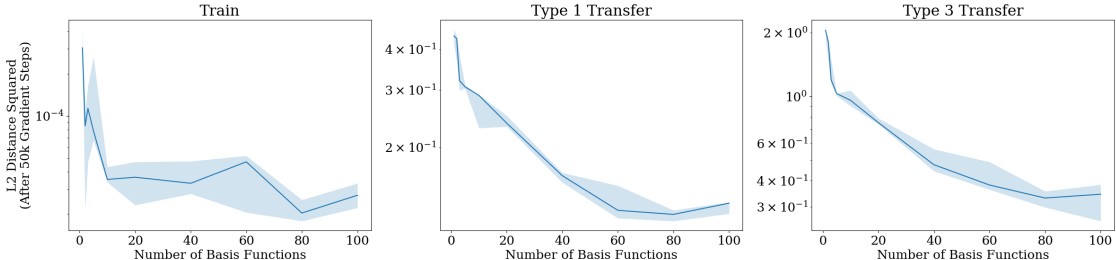

*Figure 12.* **An Ablation on the Number of Basis Functions - CIFAR Dataset.** This figure shows the final performance of FE (LS) after training for a varying number of basis functions. The figure shows steady improvement as the number of basis functions grows, until it levels off around 20 basis functions. This indicates only 20 features are necessary for good performance.

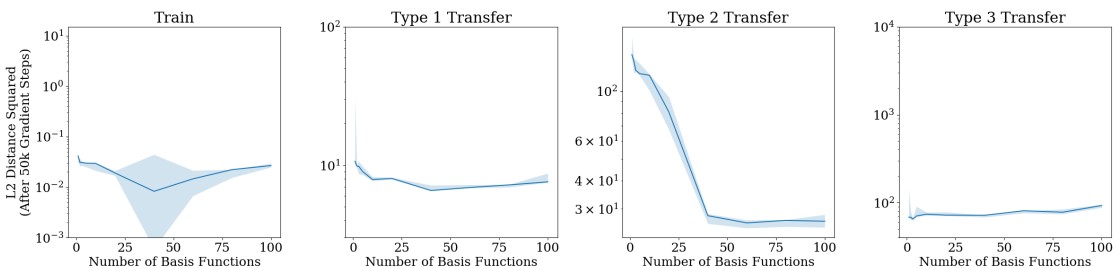

*Figure 13.* **An Ablation on the Number of Basis Functions - 7-Scenes Dataset.** This figure shows the final performance of FE (LS) after training for a varying number of basis functions. The figure shows steady improvement as the number of basis functions grows, although it begins leveling off around 100 basis functions. This indicates the dimensionality of the space is relatively high.

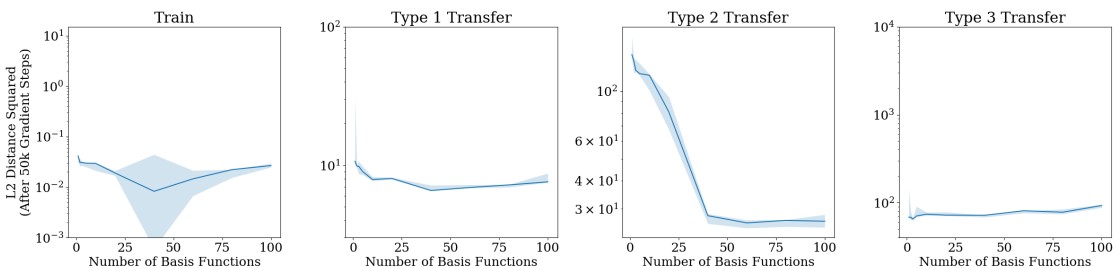

*Figure 14.* **An Ablation on the Number of Basis Functions - MuJoCo Ant Dataset.** This figure shows the final performance of FE (LS) after training for a varying number of basis functions. For type 1 and type 2 transfer, increasing the number of basis functions improves performance until around 50 basis functions, where it levels off. Interestingly, the number of basis functions does not greatly affect type 3 transfer performance, suggesting type 3 transfer in this setting is challenging.

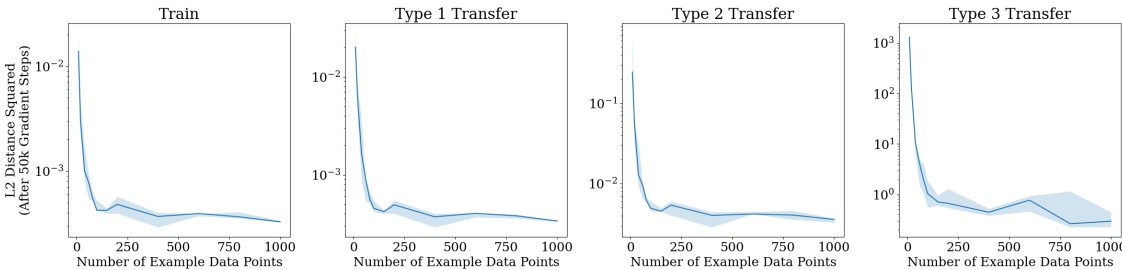

*Figure 15.* **An Ablation on the Number of Example Data Points - Polynomial Dataset.** This figures shows the final performance of FE (LS) after training for a varying number of example data points.

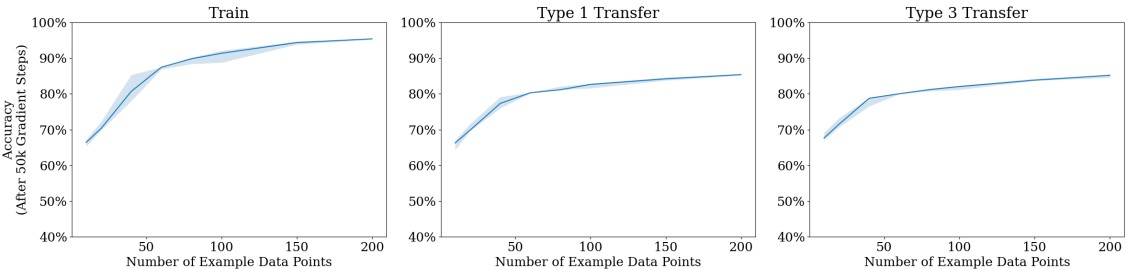

*Figure 16.* **An Ablation on the Number of Example Data Points - CIFAR Dataset.** This figures shows the final performance of FE (LS) after training for a varying number of example data points.

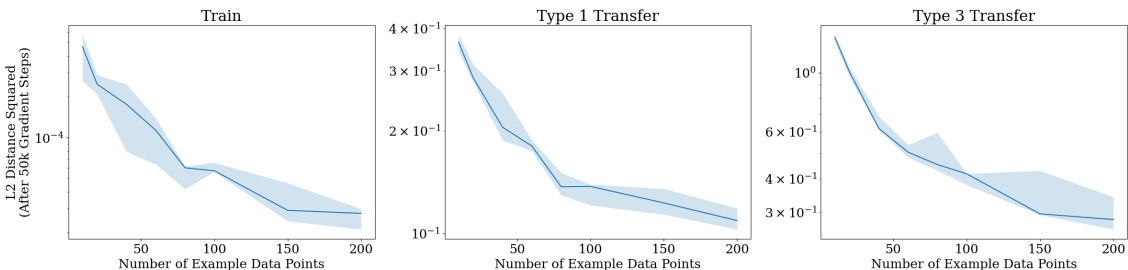

*Figure 17.* **An Ablation on the Number of Example Data Points - 7-Scenes Dataset.** This figures shows the final performance of FE (LS) after training for a varying number of example data points.

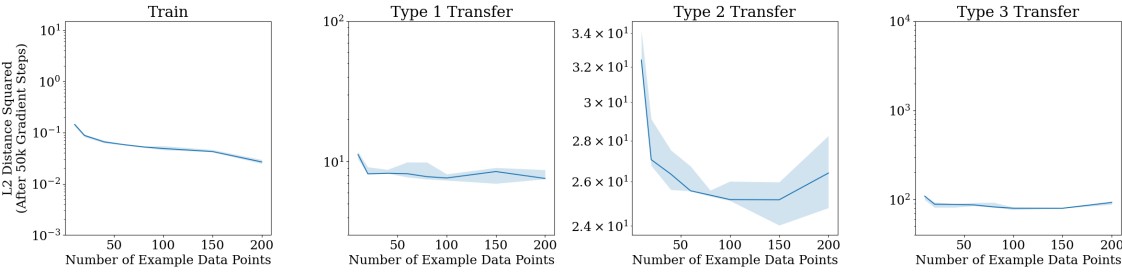

*Figure 18.* **An Ablation on the Number of Example Data Points - MuJoCo Ant Dataset.** This figures shows the final performance of FE (LS) after training for a varying number of example data points.

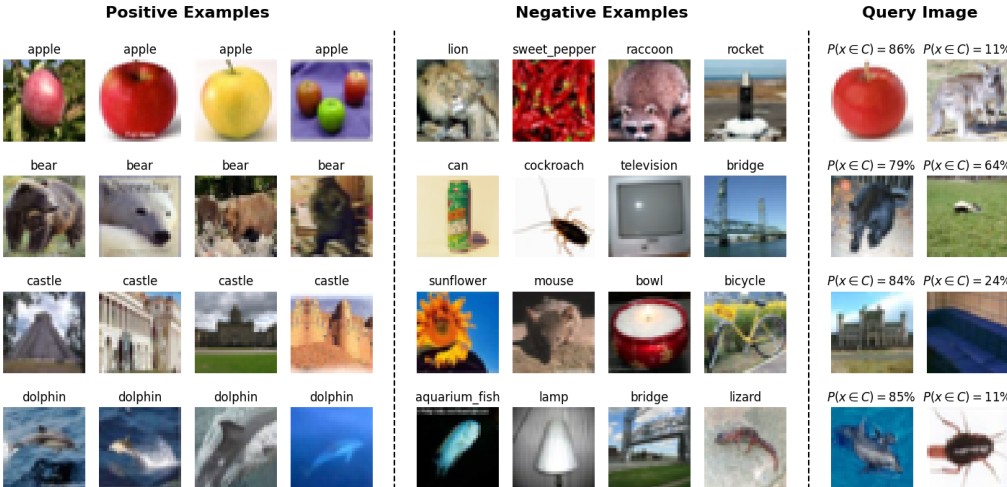

*Figure 19.* **A Visualization of the CIFAR Dataset.** For each class, 200 images are provided as examples, half of which are positive and half are negative. Positive images (left) are images from the class, whereas negative images (center) are images from *other* classes. The model should evaluate whether the query images (right) belong to the class or not. The two images on the right in each column are not evaluated together, and their probabilities are unrelated. This figure shows the performance of FE (LS) after 50,000 gradient steps in type 3 transfer. The model gets most of the shown query images correct, although it mistakes a skunk for a bear.

# H. Datasets

All figures smooth the plot with a moving average of size 40 to reduce noise in the training curves. This is necessary due to some approaches which show inconsistent performance across functions and/or input samples, especially the transformer baseline.

## H.1. CIFAR

We use a modified version of the CIFAR 100 dataset (Krizhevsky, 2009). Each function is a classifier corresponding to a specific class. For example, the "apple" classifier should label images of apples with `True` and images of anything else with `False`. 200 examples images are provided to the algorithm for each class. 100 are positive images from the correct class, and 100 are negative images from other classes, selected uniformly at random. The classifier should then classify query images as `True` if they belong to the given class and `False` otherwise. See Figure 19.

**Cross Entropy on CIFAR** We run the same CIFAR dataset with cross entropy loss functions for all baselines. We report the results in Figure 20. The results are largely unchanged, with FE (LS) and Siamese networks still performing best.

## H.2. 7-Scenes

We use a modified version of the 7-Scenes dataset (Shotton et al., 2013). We down-sample the images to 40x30 for the sake of compute time. 200 example images and their locations are provided. Then, the model should estimate the location of any new query image. We hold out the "Red Kitchen" scene for type 3 transfer. See Figure 21.

## H.3. MuJoCo Ant

We use a modified version of the MuJoCo Ant environment where the lengths of the limbs and the control authority change every episode (Ingebrand et al., 2024a). We use 10,000 episodes of length 1,000 for training. The first 200 transitions are used as example data, and the next 800 are query points. The policy is uniformly random. Type 1 hidden parameters are sampled from 0.5x to 1x the default values. Type 3 hidden parameters are sampled from 1.5x to 2x the default values. See

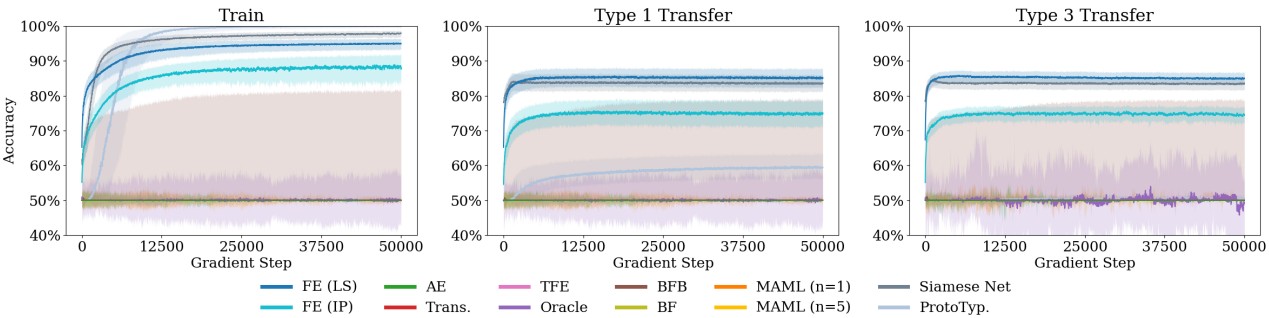

*Figure 20.* **Cross Entropy Results on CIFAR.** The empirical results for each baseline using cross entropy instead of the Hilbert space norm as a loss function. Function encoders, prototypical networks, and Siamese networks cannot use cross entropy, but are included for a comparison.

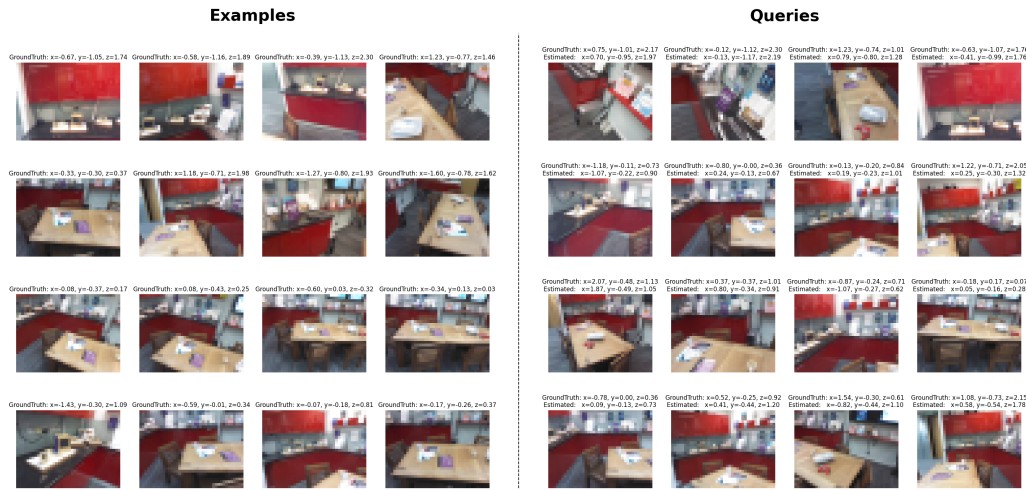

*Figure 21.* **A Visualization of the 7-Scenes Dataset.** 200 Example images (left) and their locations are provided to the model. The model then estimates the location of the query images (right). This figure shows the performance of FE (LS) in type 3 transfer.

**Type 1**        **Type 3**

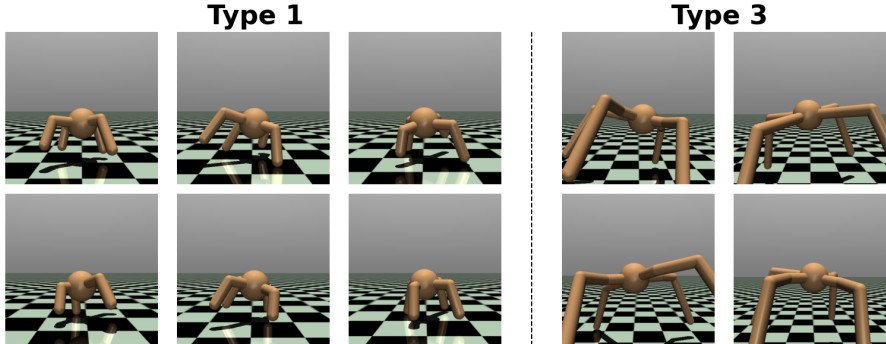

*Figure 22.* **A Visualization of the Ant Dataset.** Type 1 transfer data is generated from robots smaller than the default settings. Type 2 transfer (not shown) is linear combinations of the dynamics experienced in type 1 transfer. Type 3 transfer data is generated from robots much larger than the default settings.

Figure 22 for a visualization of how the hidden parameters affect the robot.

## I. Baselines

We implement various baselines to benchmark the function encoder against. Furthermore, we use the Hilbert space distance function as the loss function. For the non-classification datasets, this corresponds to a scaled version of mean squared error. For the classification task, we use the discrete distribution-based inner product for the main results, and cross entropy in the appendix. Below, we describe each baseline in detail. For datasets using images, an additional CNN is used to transform the image to a latent representation. As is typical, the CNN is trained as an additional component using the algorithm's standard loss function. Each algorithm is a predictor $\hat{f} : \mathcal{X} \times (\mathcal{X} \times \mathcal{Y})^m \to \mathcal{Y}$ which maps from a query point $x$ and a dataset $D_{f_T} = \{x_i, f_T(x_i)\}_{i=1}^m$ to $\hat{y}$. For most algorithms, $m$ need not be fixed. Where obvious, we omit reference to model parameters. All baselines use approximately one million total parameters.

- **FE (IP)** - The function encoder from Ingebrand et al. (2024b) which uses (4) to compute the coefficients of the basis function. This approach does not ensure orthonormality of the basis functions explicitly, but a non-orthonormal basis has high loss.

- **FE (LS)** - The modified version of the function encoder introduced in this paper. This approach uses (7) to compute the coefficients, and regularizes the basis functions to prevent exponential growth.

- **AE** - An auto encoder which has two components, the encoder and the decoder. The encoder is a learned function $e : \mathcal{X} \times \mathcal{Y} \to \mathbb{R}^k$. The encoder is used to generate a latent representation using $z := \frac{1}{m} \sum_{i=1}^m e(x_i, f(x_i))$. The decoder is a learned function $d : \mathcal{X} \times \mathbb{R}^k \to \mathcal{Y}$. The decoder is used to estimate query points using $\hat{f}(x, D_{f_T}) = d(x, z)$. The full formula is therefore $\hat{f}(x, D_{f_T}) = d(x, \frac{1}{m} \sum_{i=1}^m e(x_i, f(x_i)))$. Both components are neural networks trained on end-to-end loss.

- **Trans.** - A encoder-decoder transformer (Vaswani et al., 2017). This transformer has four components: An example encoder $e_{ex}$, a query encoder $e_q$, a transformer $t$, and a decoder $d$. The example encoder represents each example point as $z_i^{ex} := e_{ex}(x_i, f(x_i))$. The query encoder represents queries as $z^q := e_q(x)$. The transformer accepts two input sequences, one to each side of the transformer, and outputs a latent representation, $z_{out} = t(\{z_i^{ex}\}_{i=1}^m, z^q)$. Lastly, the decoder converts the latent representation to the output space, $\hat{f}(x, D_{f_T}) = d(z_{out})$. The full equation is $\hat{f}(x, D_{f_T}) = d(t(\{e_{ex}(x_i, f(x_i))\}_{i=1}^m, e_q(x)))$. All components are trained on end-to-end prediction loss. Positional encodings are not used because the data is unordered, and empirically they do not improve performance for unordered data.

- **TFE** - Transformer functional encodings. This model uses a decoder-only transformer to output the coefficients of basis functions. This model has four components: an example encoder $e$, a transformer $t$, and decoder $d$, and a set of basis functions $g$. The encoder represents the example points $z_i := e(x_i, f(x_i))$. The transformer converts the

set of example point representations to an output representation, $z_{out} := t(\{z_i\}_{i=1}^m)$. The decoder converts the latent representation to the coefficients of the basis functions, $c := d(z_{out})$. Lastly, the estimate is a traditional linear combination of basis functions, $\hat{f}(x, D_{f_T}) = g(x)^\top c$. The full formula is $\hat{f}(x, D_{f_T}) = g(x)^\top d(t(\{e(x_i, f(x_i))\}_{i=1}^m))$. All four components are trained on an end-to-end loss.

- **Oracle** - The oracle is learned function $f_\theta$ which is provided with hidden information $H$. In the Polynomial dataset, $H$ is the coefficients of the polynomials. In the CIFAR dataset, $H$ is a one-hot encoding of the class. In the 7-Scenes dataset, $H$ is a one-hot encoding of the scene (the sequence, specifically). Lastly, in the Ant dataset, $H$ is a normalized version of the hidden environment parameters (e.g. the length of the robot's limbs). Thus the estimate is $\hat{f}(x, H) = f_\theta(x, H)$, and this neural network is trained end-to-end.

- **BFB** - A naive brute force implementation with basis functions. BFB is two neural networks, a coefficient calculator $b : (\mathcal{X} \times \mathcal{Y})^m \to \mathbb{R}^k$ and basis functions $g : \mathcal{X} \to \mathcal{Y}^k$. $b$ is a neural network which takes all of the example data as input and outputs the coefficients of the basis functions, $c := b(\{x_i, f(x_i)\}_{i=1}^m)$. Thus it has the form $\hat{f}(x, D_{f_T}) = b(\{x_i, f(x_i)\}_{i=1}^m)^\top g(x)$. It is trained end-to-end.

- **BF** - A naive brute force implementation. BF is a single neural network which takes as input all of the example data and the query point, and outputs the estimated value. Thus it has the form $\hat{f}(x, D_{f_T}) = f_\theta(x, \{x_i, f(x_i)\}_{i=1}^m)$. It is trained end-to-end.

- **MAML** - A meta learning algorithm for few-shot adaptation. This algorithm consists of a single model $f_\theta : \mathcal{X} \to \mathcal{Y}$. The inference procedure consists of using the example dataset $D_{f_T}$ to fine-tune $f_\theta$, and then using this fine-tuned model to estimate the query points. This is known as the inner training step. During training, gradients are back-propagated through the inner training step to update the initial model weights $\theta$. This is know as the outer training step. MAML trains the model $f_\theta$ to be easily fine-tuned for a given source target task. An additional hyper parameter, $n$, is the number of internal gradient steps. For more information on this algorithm, see (Finn et al., 2017). MAML additionally required hyper-parameter tuning, especially the internal learning rate. We find that the best internal learning rate for one type of transfer is not typically the best for the other types of transfer, and sometimes MAML may even be unstable, as seen in the Ant experiments.

The following ad-hoc algorithms are only applicable to classification problems, in this case the CIFAR dataset.

- **Siamese networks** - Uses a triplet loss to learn a latent space which maximizes the distance between different classes. The chosen class for a query image is the label of the most similar image in the example dataset. See Bromley et al. (1993) for more information.

- **Prototypical networks** - Learns a latent representation of each image. The *prototype* of a class is the average of the representations of the corresponding images, and the label for a query image is the class corresponding to the nearest prototype in the latent space. See Snell et al. (2017) for more information.

**Connections between Siamese networks and Hilbert spaces.** Siamese networks performed well on the few-shot classification problem. We believe this is because there are deep connections between Siamese networks and Hilbert space theory. Siamese networks use a contrastive loss to minimize the distance between inputs belonging to the same category and to maximize the distance otherwise. If you consider the network as a set of basis functions, then the mean output from one category can be interpreted as a mean embedding from kernel literature. Consequently, the difference in these mean embeddings is analogous to the maximum mean discrepancy. Therefore, maximizing the distance between individual embeddings from different classes is similar to maximizing the difference between mean embeddings, and thus Siamese networks are potentially learning basis functions which maximally distinguish between the distributions corresponding to each category.

