# OpenReview forum: "Function Encoders: A Principled Approach to Transfer Learning in Hilbert Spaces"
_ICML.cc/2025/Conference — ICML 2025 poster_

### Official Review · Reviewer_PQ3a · 2025-03-14

**Overall Recommendation:** 3

**Summary:**

This work provides a geometric characterization of transfer learning in the form of 3 types of inductive transfers presented in a Hilbert space. A type 1 transfer is when the predictor of the target task is the Convex Hull of the source predictors, type 2 is when the target is the linear span of the source predictors, and type 3 is when the target sits outside the linear span.

The paper uses the theory of function encoders to propose a method that achieves transfer learning across all 3 types. This is done by providing a closed-form least-squares solution to fit the target predictor (i.e. L2 projection onto the linear span for source predictors).

 A detailed study  of the proposed method against other transfer learning benchmarks is presented, showcasing how the method can match and outperform SOTA in various settings.

**Claims And Evidence:**

The claims in the paper are mostly supported by theoretical and empirical analysis.

**Essential References Not Discussed:**

Most relevant and related work is referenced in the paper.

**Experimental Designs Or Analyses:**

The experimental domains include toy models, small-ish image tasks (CIFAR 100, 7-Scenes) and the MuJoCo Ant (dynamic control). I think the paper can benefit from additional experiments in other domains, e.g. text, structured prediction.

**Methods And Evaluation Criteria:**

IMO, L2 loss and accuracy metrics are reasonably chosen based on the setting.

**Other Comments Or Suggestions:**

Use of H for both the Hilbert space and Convex Haul (even with different fonts) can be confusing, especially since predictors live in both of them.

**Other Strengths And Weaknesses:**

The main premises of this work are interesting and insightful. The use of L2 projection onto the span of source predictors is novel to this work (to the best of my knowledge), and experiments suggest it outperforms the inner product method (Ingebrand et al., 2024b).

The main theorem in the paper is not very strong IMO, as it mainly applies in the limit of infinite basis functions.

**Questions For Authors:**

When comparing ad-hoc methods (Siamese and prototypical networks) we see comparable performance to the proposed method in multiple settings. Do you have any insights into how these ad-hoc methods achieve the type 2 or 3 transfer in the cases they can?

**Relation To Broader Scientific Literature:**

The paper compares and contrasts with prior work in great detail in section 1.1. Specifically there are comparisons with common approaches in transfer learning, meta-learning and kernel methods, in terms of sample and computational complexity and how they differ in transfer efficiency.

**Theoretical Claims:**

The proof for the theoretical claims (Theorem 1) is provided in the appendix. Insights into the design of the inner product for other spaces (e.g. probably spaces) are also provided in the appendix.

---

> ### Author Rebuttal · Authors · 2025-03-28
>
> 1. I think the paper can benefit from additional experiments in other domains, e.g. text, structured prediction.
>
> Function encoders are applicable in many domains. However, there are additional theoretical questions on how to define an appropriate inner product and geometric characterization of task relatedness for text prediction and structured representations that would need to be addressed first, and are thus beyond the scope of the current work.
>
> 2. The main theorem in the paper is not very strong IMO, as it mainly applies in the limit of infinite basis functions.
>
> Please see our response to reviewer qZy4, question 2. Universal approximation theorems are existence proofs, meaning they provide justification that the proposed approach could achieve arbitrary accuracy given enough resources. Practically, Theorem 1 guarantees the existence of a function encoder which spans the source tasks, which is necessary for the types of transfer we characterize.
>
> 3. Use of H for both the Hilbert space and Convex Haul (even with different fonts) can be confusing, especially since predictors live in both of them.
>
> Thank you for this comment, we will change the notation for convex hull in the final version.
>
> 4. When comparing ad-hoc methods (Siamese and prototypical networks) we see comparable performance to the proposed method in multiple settings. Do you have any insights into how these ad-hoc methods achieve the type 2 or 3 transfer in the cases they can?
>
> Siamese networks use a contrastive loss to minimize the distance between inputs belonging to the same category and to maximize the distance otherwise. We believe there are deep theoretical connections between this approach and maximum mean discrepancy (MMD) from kernel methods, where you could view the mean output across a category as a mean embedding, and the distance between embeddings as the MMD. Therefore, maximizing the distance between individual embeddings is somehow similar to maximizing the difference between mean embeddings, and thus Siamese networks are effectively learning basis functions which maximally distinguish between the distributions corresponding to each category. Prototypical networks do something similar, but instead work with the mean embedding directly, using the closest mean embedding to a given input as the correct category. Therefore we believe these approaches have unrecognized theoretical connections to Hilbert space theory, which potentially explains their performance.

---

> > ### Comment · Reviewer_PQ3a · 2025-04-04
> >
> > Thanks for the comments and explanations. I suggest adding the discussion around ad-hoc methods to the appendix or even the main body of the paper. It would indeed be interesting to analyse (in some future work) if these methods are implicitly building basis functions.
> >
> > The existence proof is useful and needed for the discussion around typing, but it does not provide a finite construction path or convergence rate. This IMO limits the significance of the theoretical analysis provide in the work.

---

### Official Review · Reviewer_qZy4 · 2025-03-23

**Overall Recommendation:** 2

**Summary:**

This paper studies inductive transfer learning where the source tasks and the target task share the same domain $(\mathcal{X}, \mathbb{P}(\mathcal{X}))$ and output space $\mathcal{Y}$, but have different predictors $f:\mathcal{X}\to\mathcal{Y}$ that lie in some Hilbert space. It characterizes the difficulty of transfer by the geometry of the source and target predictors. Drawing inspirations from prior works on function encoders, it proposes to learn a set of basis functions that approximately span the Hilbert space of predictors, so as to adapt to future tasks without retraining the model. To train function encoders, it applies least squares which is more efficient than the previously used inner product method. Numerical experiments on polynomial regression, image binary classification, position estimation from image, and robot state prediction demonstrate the effectiveness of the algorithm.

## Update after Rebuttal
I thank the authors for the detailed response. I still think that the work [1] diminishes the novelty of the methodology of this work. For the geometric view of transfer, the authors explained with the example of classifying horses and classifying lung cancer, but I believe that one can tell the two tasks are highly dissimilar without the Hilbert space framework. The geometric categorization, while intuitive, does not seem to provide any fundamentally deeper understanding of the problem, and does not really give insights into the development of the methodology. Therefore, given the limited novelty of the theoretical framework and the methodology, I choose to maintain my score.

**Claims And Evidence:**

The claims made in the paper are supported by clear and convincing evidence.

**Essential References Not Discussed:**

A key methodological contribution of this paper is to replace the inner product method in the function encoder training procedure by the least squares method. However, it seems that the least squares method is not new, as it already exists in the paper [1] which proposes the same method for the same purpose of overcoming the orthogonality assumption. (See, e.g., Figure 2 in that paper.)

**Reference**

[1] Ingebrand et al. (2024). Basis-to-basis operator learning using function encoders. https://doi.org/10.1016/j.cma.2024.117646

**Experimental Designs Or Analyses:**

The experiment procedures and results in the paper are clearly described, and the results look convincing.

**Methods And Evaluation Criteria:**

The proposed method is conceptually simple and intuitive. It is tested on four different datasets covering different applications.

**Other Comments Or Suggestions:**

The notations for domain ($\mathcal{D}$) and dataset ($D$) use the same letter "D", which can be a bit confusing.

**Other Strengths And Weaknesses:**

**Strengths:**

1. The paper is well written, and the ideas are clearly explained.

2. The proposed method is conceptually simple and intuitive.


**Weaknesses:**

1. Originality of the proposed approach. The least squares method for function encoder training seems to already exist in the literature. See "Essential References Not Discussed" for more details.

2. Significance of Theorem 1. While the theorem is correct, I feel that it is mathematically trivial given the universal approximation theorem of neural networks. In particular, the result is qualitative; it only states that any predictor in the Hilbert space can be approximated using sufficiently many basis functions. More importantly, it offers minimal insights into the transfer learning problem. Indeed, the basis functions are learned using source tasks, and whether the basis functions can well approximate the target task should depend on the similarity and the diversity of the tasks. This important problem is not covered by Theorem 1.

3. Implications of the geometric view of transfer. The paper divides the transfer learning problem into three types, according to whether the target predictor lies in the convex hull or the linear span of the source predictors. The geometric characterization is intuitive, and the numerical experiments confirm the three types of problems have different difficulty levels. However, beyond these points, I do not see how the geometric view can give us more insights into the difficulty of the transfer learning problem, or how it can used to facilitate transfer learning in specific scenarios.

**Questions For Authors:**

1. Ablation studies in the paper show that the choice of the number $k$ of basis functions can have a significant impact on the performance of the algorithm. However, in practice, the appropriate $k$ is unknown beforehand. While the experiments suggest that $k$ can be chosen as the dimensionality of the space, it can be too large and make the algorithm computationally expensive. Do the authors have a principled way of choosing an appropriate $k$?

**Relation To Broader Scientific Literature:**

This paper proposes a general approach for transfer learning and demonstrates its effectiveness for a variety of tasks. Transfer learning is still a widely open problem in machine learning, and is important for various applications including image classification and robotics.

**Theoretical Claims:**

I have checked that Theorem 1 and its proof are correct.

---

> ### Author Rebuttal · Authors · 2025-03-28
>
> 1. Originality of the proposed approach. The least squares method for function encoder training seems to already exist in the literature. See "Essential References Not Discussed" for more details.
>
> We thank the reviewer for pointing out the referenced work in [1], and we have added the reference to our final manuscript. To clarify the originality of our contribution in relation to [1], the development of the referenced work occurred concurrently with our manuscript. Indeed, the referenced work [1] uses the least-squares formulation and implementation from this work, despite appearing first. Leaving out [1] as a reference was an oversight and has been corrected.
>
> In any case, while [1] addresses operator learning and PDE modeling, this paper is concerned with function space approximation. We introduce a geometric characterization of transfer, we  define the necessary regularization for the LS approach to converge, we generalize the inner product definition, we provide a universal function space approximation theorem, and we compare against SOTA baselines such as meta learning and transformers on various benchmarks, including a classification task that function encoders have not previously been able to handle. Thus while both approaches use LS, these two papers are distinct.
>
> 2. Significance of Theorem 1. While the theorem is correct, I feel that it is mathematically trivial given the universal approximation theorem of neural networks...
>
> We agree that, in general, universal approximation theorems are fundamentally existence proofs. However, they play a foundational role in justifying neural networks as universal approximators of continuous functions, and thus have significant practical value. Nevertheless, we agree that universal approximation theorems may have limited use in determining the quantitative computational requirements or error bounds, which vary from problem to problem and depend on the training data.
>
> 3. More importantly, it offers minimal insights into the transfer learning problem. Indeed, the basis functions are learned using source tasks, and whether the basis functions can well approximate the target task should depend on the similarity and the diversity of the tasks. This important problem is not covered by Theorem 1.
>
> From our geometric categorization of transfer, we know that the difficulty of approximating a new task depends on its relation to the source tasks, e.g. whether it lies in the span of the source tasks and if not, the distance from the span of the source task as measured by the Hilbert space norm. On the other hand, since the function encoder trains basis functions, its approximation error is proportional to the distance between the span of the basis and the target task. Therefore, the approximation error of a function encoder on a new task aligns with the expected difficulty of that target task according to the geometric categorization of transfer. UAT is useful because it guarantees the existence of a function encoder which spans the source tasks. So assuming that the training procedure converges, we expect the function encoder’s approximation error to be conceptually equivalent to the difficulty of transfer suggested by the geometric view. We have added a paragraph highlighting this connection at the end of section 3.
>
> 4. I do not see how the geometric view can give us more insights into the difficulty of the transfer learning problem, or how it can used to facilitate transfer learning in specific scenarios.
>
> This is a great question. The geometric view of transfer provides a framework for categorizing transfer problems. Consider classifying images by attributes, such as identifying horses or stripes. If you want to detect zebras (striped horses), this likely involves type 1 transfer, where existing methods typically perform well. However, then classifying lung cancer from X-rays isn’t a linear combination of previous attributes, indicating type 3 transfer, making it significantly harder despite the apparent similarity between tasks. Thus, the geometric view helps assess the difficulty of transfer problems and suggests suitable approaches; for instance, our results indicate transformers aren’t effective for type 3 transfer.
>
> 5. Do the authors have a principled way of choosing an appropriate k?
>
> Please see our response to reviewer wRbf for questions 1 and 2 for additional details. The overhead for choosing a larger k is relatively small.
>
> Furthermore, the ablations show that k=100 is enough for all of the problems in this paper; This aligns with prior work. If k is chosen to be less than the dimensionality of the space, then the algorithm will likely learn the k principle basis functions, since doing so would minimize the loss. For example, see the ablation on k for the CIFAR dataset, where most of the prediction accuracy is due to 20 basis functions, and using additional basis functions provides only diminishing returns.

---

> > ### Comment · Reviewer_qZy4 · 2025-04-04
> >
> > I thank the authors for the detailed response. I still think that the work [1] diminishes the novelty of the methodology of this work. For the geometric view of transfer, the authors explained with the example of classifying horses and classifying lung cancer, but I believe that one can tell the two tasks are highly dissimilar without the Hilbert space framework. The geometric categorization, while intuitive, does not seem to provide any fundamentally deeper understanding of the problem, and does not really give insights into the development of the methodology. Therefore, given the limited novelty of the theoretical framework and the methodology, I choose to maintain my score.

---

### Official Review · Reviewer_wRbf · 2025-03-23

**Overall Recommendation:** 4

**Summary:**

The authors define three types of inductive transfer in Hilbert spaces: interpolation within the convex hull, extrapolation to the linear span, and extrapolation outside the span. They propose to learn neural network basis functions, named as function encoders, that can represent any function in this space. Specifically, they employ a least-squares formulation to compute the coefficients of the basis functions. The paper support their method with a universal function space approximation theorem, showing that any function in a separable Hilbert space can be approximated arbitrarily well if enough basis functions are used. They validate their approach through experiments on several benchmarks, including polynomial regression, CIFAR image classification, camera pose estimation, and dynamics estimation on robotic data.

## update after rebuttal
I appreciate the authors' detailed response and the additional experiments addressing the computational cost of increasing $k$, as well as the discussion on basis function regularization and domain shifts. Based on these clarifications, I will maintain my overall score.

**Claims And Evidence:**

The paper’s claims are well supported by both theoretical analysis and experimental results. I did not encounter any obviously problematic or unclear statements in the claims.

**Essential References Not Discussed:**

No.

**Experimental Designs Or Analyses:**

I checked the Section 4 in the paper. Overall, the experimental designs are sound and cover a diverse set of tasks. However, addressing the points above could further solidify the claims and help clarify the practical limitations of the proposed approach.

**Methods And Evaluation Criteria:**

Yes. The authors evaluate their approach on a wide range of datasets, from simulated polynomial regression to real-world tasks like CIFAR image classification, camera pose estimation on the 7-Scenes dataset, and dynamics estimation with MuJoCo. This diversity in experiments demonstrates the versatility of the method across different transfer learning scenarios, especially in cases requiring extrapolation. Some issues are noted in the question section.

**Other Comments Or Suggestions:**

1. I would suggest some additional experiments on the computational cost of increasing $k$. It would be better if there are some experiments regarding a reasonable method for choosing $k$ from the dataset adaptively.
2. It would be beneficial to explicitly write out the details of Section A Basis Function Regularization, e.g., the form of additional regularizer on the diagonal of Gram matrix.

**Other Strengths And Weaknesses:**

The paper presents a geometric perspective for transfer learning, distinguishing between different types of transfer. Overall, the manuscript is well-structured, making it easy to follow the progression from theoretical motivation to algorithmic details and experimental validation. The experimental comparisons are comprehensiv, covering a broad spectrum of tasks, enhancing method's credibility.
See Questions part for weakness.

**Questions For Authors:**

1. The method assumes that the multiple source and target domains are identical, i.e., $D_{S_1}=\cdots=D_{S_n}=D_T$, which is a strong assumption in practice. Given that many real-world scenarios involve some degree of domain shift or weak task relatedness, what specific aspects of the FE framework could be adapted to handle cases when there exists some $j$ such that $D_{S_j}\ne D_T$? It would be beneficial to discuss potential extensions to their framework under domain shifts.

2. Given that Section G suggests that the optimal $k$ varies across datasets, what is the impact of larger $k$ on computational cost across all tasks? Is there a reasonable method to balance empirical performance with computational efficiency?

3. How does learning neural network–based basis functions provide advantages over fixed basis methods (e.g., kernel method or dictionary atoms) in capturing complex nonlinearities and enabling robust extrapolation, particularly for Type 2 and Type 3 transfer? It would be helpful to discuss the limitations in fixed basis methods and explain how a learned representation can adapt to more intricate data structures in challenging transfer scenarios.

**Relation To Broader Scientific Literature:**

The paper’s contributions are closely connected to kernel methods. The proposed approach shares similar spirit to that in kernel learning, where functions are represented in a RKHS, but here the authors replace fixed kernels with learned neural network basis functions. This provides a more flexible representation that can adapt to complex data structures.

**Theoretical Claims:**

I checked the detailed proof provided for Theorem 1, which establishes a universal approximation guarantee for the function encoder framework. Overall, the proof is correct in its high-level idea and is built upon classical universal approximation results.

---

> ### Author Rebuttal · Authors · 2025-03-28
>
> 1. I would suggest some additional experiments on the computational cost of increasing k.
>
> Using the worst-case strategy, the compute time for a forward pass is linear in k if you compute the basis functions sequentially on a single thread. Beyond the cost for the forward pass, the compute time for the Gram matrix inverse is cubic in k, but is typically insignificant relative to the forward pass.
>
> However, in practice, the compute time for a forward pass is approximately constant with respect to the number of basis functions since each can be run simultaneously via GPU parallelization. Therefore, there is little overhead for increasing k. Empirically, the percentage increase in training time of k=100 relative to k=1 is the following:
>
> Polynomial - 14.4 %
>
> CIFAR - 8 %
>
> 7 Scenes - 11.5 %
>
> Ant - 17.8 %
>
> Indeed, the overhead is minor even though we increase k by two orders of magnitude. We will include a graph comparing training time for different numbers of basis functions in the appendix.
>
> 2. It would be better if there are some experiments regarding a reasonable method for choosing k from the dataset adaptively.
>
> We agree that exploring adaptive methods to reduce the number of basis functions (either during training or after training) is an interesting area for future research. We anticipate that L1 regularization can be used to impose sparsity during training, or that we could "constructively" train a set of basis functions--adding new basis functions periodically until we encounter diminishing returns. However, in most cases the overhead of choosing a large k is sufficiently small such that often the best solution is to simply choose a large value of k at the start.
>
> 3. It would be beneficial to explicitly write out the details of Section A Basis Function Regularization, e.g., the form of additional regularizer on the diagonal of Gram matrix.
>
> We will improve the discussion of regularization in the final version by incorporating the regularization loss shown in Algorithm 1 as part of the main text.
>
> 4. ... It would be beneficial to discuss potential extensions to their framework under domain shifts.
>
> This is an excellent point. In practice, we only require that the input space X is the same for all datasets. However, the distribution between tasks can be different. If the distributions are entirely disjoint, this is a special case of the weighted inner product described in Appendix D. So the algorithm is still applicable, but implicitly the inner product’s definition changes between domains. Note that both image-based experiments in this paper include domain transfer in addition to function space transfer, since the distribution of images is different for different classes/scenes. Implicitly, our use of least squares calibrates the function estimate to new domains. We will improve the discussion in Appendix D to highlight these details.
> Furthermore, analyzing the interplay between domain shifts, inner products, and least squares is an interesting direction for future work.
>
> 5. Given that Section G suggests that the optimal k varies across datasets, what is the impact of larger k on computational cost across all tasks? Is there a reasonable method to balance empirical performance with computational efficiency?
>
> Please see our response to question 1 above for the computational cost of choosing k and the scaling across tasks. However, increasing k does not reduce prediction accuracy. Therefore, a good naive strategy is to simply choose large k, which maximizes performance while only mildly increasing compute time. As mentioned in our response to question 2 above, we anticipate that we could "constructively" train a set of basis functions if computational efficiency is critical.
>
>
> 6. How does learning neural network–based basis functions provide advantages over fixed basis methods... It would be helpful to discuss the limitations in fixed basis methods and explain how a learned representation can adapt to more intricate data structures in challenging transfer scenarios.
>
> We anticipate that kernel methods and dictionary learning are amenable to type 2 & 3 transfer due to the use of least squares. However, the key advantage of function encoders over kernel methods is that the Gram matrix inverse in kernel methods scales cubically with the amount of data, whereas the scaling is cubic with the hyper-parameter k for function encoders.  Additionally, we note that function encoders avoid the issue of pre-specifying a kernel, and therefore may learn basis functions which are specialized to the given problem. On the other hand, dictionary learning is effectively learning a discretized version of basis functions evaluated at a fixed mesh of input locations, and so cannot be queried at new points, which makes it intractable for domains like robotics where we can’t measure the same state twice. In contrast, the function encoder is queryable at any new point due to the use of neural networks.

---

### Decision · Program_Chairs · 2025-05-01

**Decision:**

Accept (poster)

**Comment:**

This paper presents a geometric framework for transfer learning in Hilbert spaces, identifying three transfer types: interpolation, extrapolation within span, and beyond. The authors propose a function encoder trained via least-squares, backed by a universal approximation theorem. Experiments show it outperforms transformers and meta-learning across diverse benchmarks and transfer scenarios. The reviewers appreciate the simplicity and clarity of the proposed method. They also pointed out a few limitations in the theory and experiment. The authors should revise the paper according to the reviews and rebuttals.